# RatInABox, a toolkit for modelling locomotion and neuronal activity in continuous environments

**Tom M George[1]\*, Mehul Rastogi[1], William de Cothi[2†], Claudia Clopath[1,3†], Kimberly Stachenfeld[4,5†], Caswell Barry[2†]**

[1]Sainsbury Wellcome Centre, University College London, London, United Kingdom; [2]Department of Cell and Developmental Biology, University College London, London, United Kingdom; [3]Department of Bioengineering, Imperial College London, London, United Kingdom; [4]Google DeepMind, London, United Kingdom; [5]Columbia University, New York, United States

**Abstract** Generating synthetic locomotory and neural data is a useful yet cumbersome step commonly required to study theoretical models of the brain's role in spatial navigation. This process can be time consuming and, without a common framework, makes it difficult to reproduce or compare studies which each generate test data in different ways. In response, we present RatInABox, an open-source Python toolkit designed to model realistic rodent locomotion and generate synthetic neural data from spatially modulated cell types. This software provides users with (i) the ability to construct one- or two-dimensional environments with configurable barriers and visual cues, (ii) a physically realistic random motion model fitted to experimental data, (iii) rapid online calculation of neural data for many of the known self-location or velocity selective cell types in the hippocampal formation (including place cells, grid cells, boundary vector cells, head direction cells) and (iv) a framework for constructing custom cell types, multi-layer network models and data- or policy-controlled motion trajectories. The motion and neural models are spatially and temporally continuous as well as topographically sensitive to boundary conditions and walls. We demonstrate that out-of-the-box parameter settings replicate many aspects of rodent foraging behaviour such as velocity statistics and the tendency of rodents to over-explore walls. Numerous tutorial scripts are provided, including examples where RatInABox is used for decoding position from neural data or to solve a navigational reinforcement learning task. We hope this tool will significantly streamline computational research into the brain's role in navigation.

**\*For correspondence:**
tom.george.20@ucl.ac.uk

†Joint lead authors

**Competing interest:** The authors declare that no competing interests exist.

## Editor's evaluation

RatInABox is a new python library for generating synthetic behavioral and neural data (many functional cell types) that is a highly important contribution to computational neuroscience. Critically, the authors have gone beyond the generally accepted practice with their well-written paper, documented and verified code. They show compelling evidence of its utility and usability, and this is sure to be an influential paper with implications for developing new theories and methods for joint neural and behavioral analysis beyond the navigation field.

**eLife digest** The brain is a complex system made up of over 100 billion neurons that interact to give rise to all sorts of behaviours. To understand how neural interactions enable distinct behaviours, neuroscientists often build computational models that can reproduce some of the interactions and behaviours observed in the brain.

Unfortunately, good computational models can be hard to build, and it can be wasteful for different groups of scientists to each write their own software to model a similar system. Instead, it is more effective for scientists to share their code so that different models can be quickly built from an identical set of core elements. These toolkits should be well made, free and easy to use.

One of the largest fields within neuroscience and machine learning concerns navigation: how does an organism – or an artificial agent – know where they are and how to get where they are going next? Scientists have identified many different types of neurons in the brain that are important for navigation. For example, 'place cells' fire whenever the animal is at a specific location, and 'head direction cells' fire when the animal's head is pointed in a particular direction. These and other neurons interact to support navigational behaviours.

Despite the importance of navigation, no single computational toolkit existed to model these behaviours and neural circuits. To fill this gap, George et al. developed RatInABox, a toolkit that contains the building blocks needed to study the brain's role in navigation. One module, called the 'Environment', contains code for making arenas of arbitrary shapes. A second module contains code describing how organisms or 'Agents' move around the arena and interact with walls, objects, and other agents. A final module, called 'Neurons', contains code that reproduces the reponse patterns of well-known cell types involved in navigation. This module also has code for more generic, trainable neurons that can be used to model how machines and organisms learn.

Environments, Agents and Neurons can be combined and modified in many ways, allowing users to rapidly construct complex models and generate artificial datasets. A diversity of tutorials, including how the package can be used for reinforcement learning (the study of how agents learn optimal motions) are provided.

RatInABox will benefit many researchers interested in neuroscience and machine learning. It is particularly well positioned to bridge the gap between these two fields and drive a more brain-inspired approach to machine learning. RatInABox's userbase is fast growing, and it is quickly becoming one of the core computational tools used by scientists to understand the brain and navigation. Additionally, its ease of use and visual clarity means that it can be used as an accessible teaching tool for learning about spatial representations and navigation.

## Introduction

Computational modelling provides a means to understand how neural circuits represent the world and influence behaviour, interfacing between experiment and theory to express and test how information is processed in the brain. Such models have been central to understanding a range of neural mechanisms, from action potentials (*Hodgkin and Huxley, 1952*) and synaptic transmission between neurons (*del Castillo and Katz, 1954*), to how neurons represent space and guide complex behaviour (*Hartley et al., 2000*; *Hartley et al., 2004*; *Byrne et al., 2007*; *Banino et al., 2018*; *de Cothi et al., 2022*). Relative to empirical approaches, models can offer considerable advantages, providing a means to generate large amounts of data quickly with limited physical resources, and are a precise means to test and communicate complex hypotheses. To fully realise these benefits, computational modelling must be accessible and standardised, something which has not always been the case.

Spurred on by the proposition of a 'cognitive map' (*Tolman and Honzik, 1930*), and the discovery of neurons with position-(*O'Keefe and Dostrovsky, 1971*), velocity-(*Sargolini et al., 2006*; *Kropff et al., 2015*) and head direction-(*Taube et al., 1990*) selective receptive fields in the hippocampal formation, understanding the brain's role in navigation and spatial memory has been a key goal of the neuroscience, cognitive science, and psychology communities. In this field, it is common for theoretical or computational models to rely on artificially generated data sets. For example, for the direct testing of a normative model, or to feed a learning algorithm with training data from a motion model used to generate a time series of states, or feature-vectors. Not only is this data more cost-effective,

quicker to acquire, and less resource-intensive than conducting spatial experiments (no rats required), but it also offers the advantage of being flexibly hand-designed to support the validation or refutation of theoretical propositions. Indeed, many past (*Mehta et al., 2000*; *Burak et al., 2009*; *Gustafson and Daw, 2011*) and recent (*Stachenfeld et al., 2017*; *de Cothi and Barry, 2020*; *Bono et al., 2023*; *George et al., 2022*; *Banino et al., 2018*; *Schaeffer et al., 2022*; *Benna and Fusi, 2021*) models have relied on artificially generated movement trajectories and neural data.

Artificially generating data can still be a bottleneck in the scientific process. We observe a number of issues: First, the lack of a universal standard for trajectory and cell activity modelling hinders apples-to-apples comparisons between theoretical models whose conclusions may differ depending on the specifics of the models being used. Secondly, researchers must begin each project reinventing the wheel, writing software capable of generating pseudo-realistic trajectories and neural data before the more interesting theoretical work can begin. Thirdly, inefficiently written software can significantly slow down simulation time or, worse, push users to seek solutions which are more complex and power-intensive (multithreading, GPUs, etc.) than the underlying task requires, decreasing reproducibility. Finally, even the relatively modest complexities of motion modelling in continuous environments raises the technical entry barrier to computational research and can impel researchers towards studying only one-dimensional environments or biologically unrealistic 'gridworlds' with tabularised state spaces.

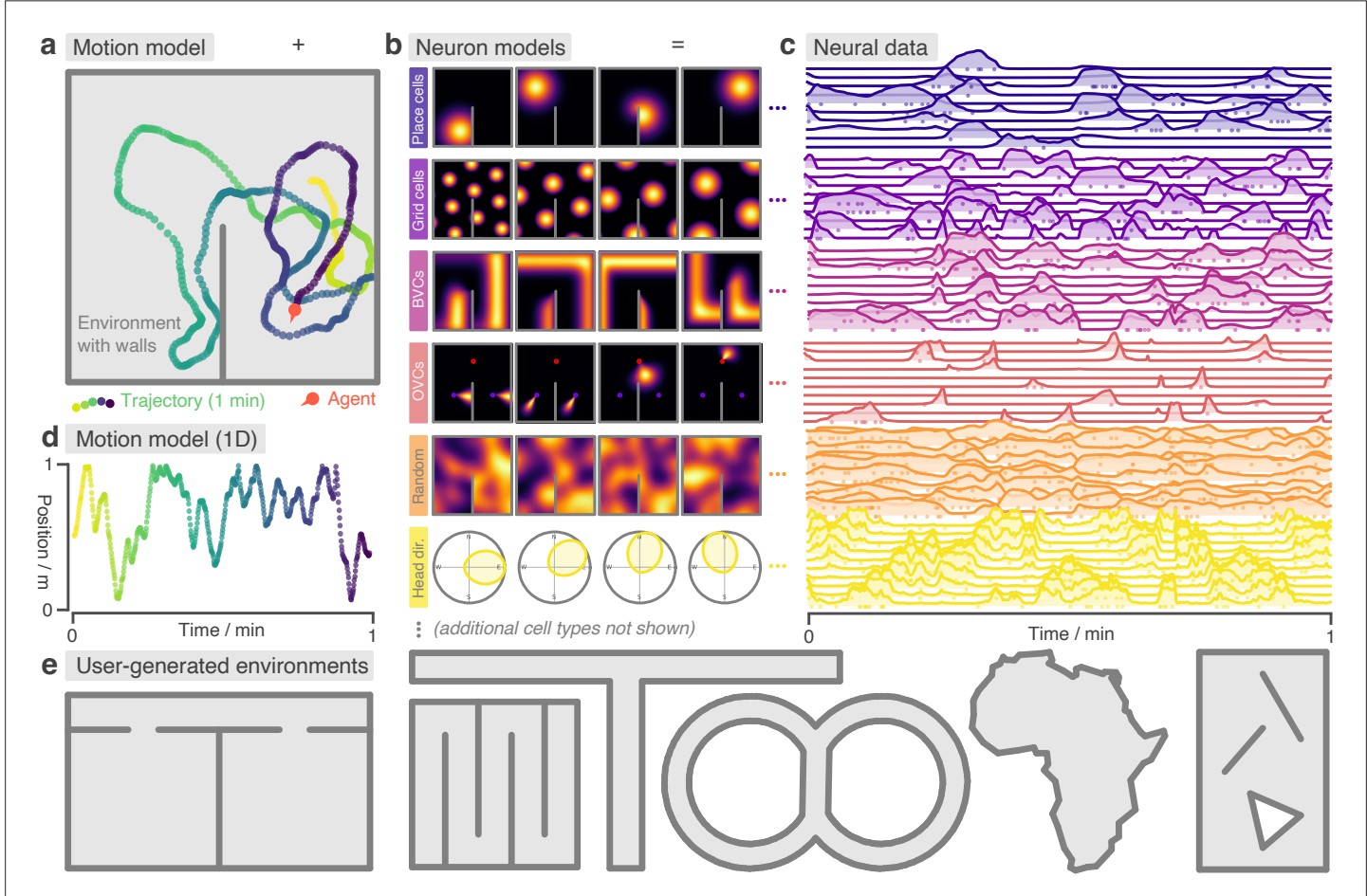

**Figure 1.** RatInABox is a flexible toolkit for simulating locomotion and neural data in complex continuous environments. (**a**) One minute of motion in a 2D `Environment` with a wall. By default the `Agent` follows a physically realistic random motion model fitted to experimental data. (**b**) Premade neuron models include the most commonly observed position/velocity selective cells types (6 of which are displayed here). Users can also build more complex cell classes based on these primitives. Receptive fields interact appropriately with walls and boundary conditions. (**c**) As the `Agent` explores the `Environment`, `Neurons` generate neural data. This can be extracted for downstream analysis or visualised using in-built plotting functions. Solid lines show firing rates, and dots show sampled spikes. (**d**) One minute of random motion in a 1D environment with solid boundary conditions. (**e**) Users can easily construct complex `Environments` by defining boundaries and placing walls, holes and objects. Six example `Environments`, some chosen to replicate classic experimental set-ups, are shown here.

Not only can gridworld models scale poorly in large environments but they typically disregard aspects of motion which can be non-trivial, for example speed variability and inertia. Whilst there are valid reasons why gridworld and/or tabularised state-space models may be preferred – and good open source packages for modelling this (*Maxime et al., 2023*; *Juliani et al., 2022*) – we suspect that coding simplicity, rather than theory-based justifications, remain a common reason these are used over continuous analogs.

To overcome these issues we built RatInABox (https://github.com/RatInABox-Lab/RatInABox) (*George, 2022*): an open source Python toolkit for efficient and realistic motion modelling in complex continuous environments *and* concurrent simulation of neuronal activity data for many cell types including those typically found in the hippocampal formation (*Figure 1*).

## RatInABox

RatInABox is an open source software package comprising three component classes:

- `Environment`: The environment (or 'box') that the `Agent` exists in. An `Environment` can be one- or two-dimensional, contain walls/barriers, holes, and objects and they can have periodic or solid boundary conditions (*Figure 1a, b, d, e*).
- `Agent`: The agent (or 'rat') moving around the `Environment` (*Figure 1a, d*). `Agent`s are 0-dimensional and `Environment`s can contain multiple `Agent`s simultaneously.
- `Neurons`: A population of neurons whose firing rates update to encode the 'state' of the `Agent` in a rich variety of ways. Specific subclasses are provided corresponding to commonly studied cell-types (including, but not limited to, `PlaceCells`, `GridCells`, `BoundaryVectorCells` and `HeadDirectionCells`, *Figure 1b, c*). Users can also write their own `Neurons` subclasses or build/train complex function-approximator `Neurons` based on these primitives.

A typical workflow would be as follows: Firstly, an `Environment` is initialised with parameters specifying its dimensionality, size, shape and boundary conditions. Walls, holes and objects (which act as 'visual cues') can be added to make the `Environment` more complex. Secondly, an `Agent` is initialised with parameters specifying the characteristics of its motion (mean/standard deviation of its speed and rotational velocity, as well as behaviour near walls). Thirdly, populations of `Neurons` are initialised with parameters specifying their characteristics (number of cells, receptive field parameters, maximum firing rates etc.).

Next, a period of simulated motion occurs: on each step the `Agent` updates its position and velocity within the `Environment`, given the duration of the step, and `Neurons` update their firing rates to reflect the new state of the `Agent`. After each step, data (timestamps, position, velocities, firing rates and spikes sampled according to an inhomogenous Poisson process) are saved into their respective classes for later analysis, *Figure 1*.

RatInABox is fundamentally continuous in space and time. Position and velocity are never discretised but are instead stored as continuous values and used to determine cell activity online, as exploration occurs. This differs from other models which are either discrete (e.g. 'gridworld' or Markov decision processes) (*Maxime et al., 2023*; *Juliani et al., 2022*) or approximate continuous rate maps using a cached list of rates precalculated on a discretised grid of locations (*de Cothi and Barry, 2020*). Modelling time and space continuously more accurately reflects real-world physics, making simulations smooth and amenable to fast or dynamic neural processes which are not well accommodated by discretised motion simulators. Despite this, RatInABox is still fast; to simulate 100 `PlaceCells` for 10 min of random 2D motion (dt = 0.1 s) it takes about 2 s on a consumer grade CPU laptop (or 7 s for boundary vector cells).

By default the `Agent` follows a temporally continuous smooth random motion model, closely matched to the statistics of rodent foraging in an open field (*Sargolini et al., 2006*, *Figure 2*); however, functionality is also provided for non-random velocity control via a user provided control signal or for the `Agent` to follow an imported trajectory (*Figure 3a*). Once generated, data can be plotted using in-built plotting functions (which cover most of the figures in this manuscript) or extracted to be used in the theoretical model being constructed by the user.

### Intended use-cases

RatInABox can be used whenever locomotion and/or populations of cells need to be modelled in continuous one- or two-dimensional environments. These functionalities are coupled (locomotion

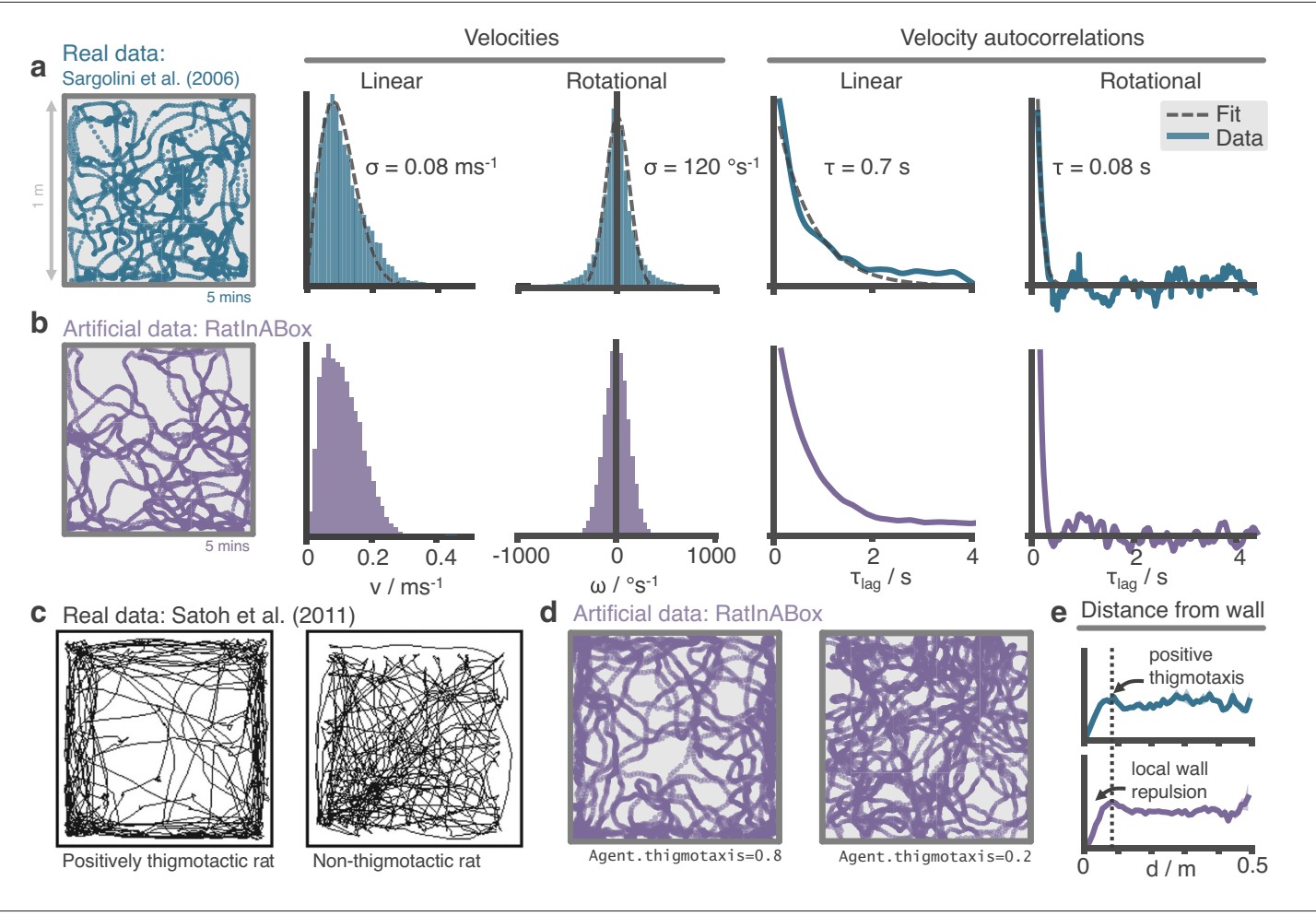

**Figure 2.** The RatInABox random motion model closely matches features of real rat locomotion. (**a**) An example 5-min trajectory from the *Sargolini et al., 2006*. dataset. Linear velocity (Rayleigh fit) and rotational velocity (Gaussian fit) histograms and the temporal autocorrelations (exponential fit) of their time series'. (**b**) A sampled 5-min trajectory from the RatInABox motion model with parameters matched to the Sargolini data. (**c**) Figure reproduced from Figure 8D in *Satoh et al., 2011* showing 10 min of open-field exploration. 'Thigmotaxis' is the tendency of rodents to over-explore near boundaries/walls and has been linked to anxiety. (**d**) RatInABox replicates the tendency of agents to over-explore walls and corners, flexibly controlled with a 'thigmotaxis' parameter. (**e**) Histogram of the area-normalised time spent in annuli at increasing distances, $d$, from the wall. RatInABox and real data are closely matched in their tendency to over-explore locations near walls without getting too close.

directly adjusts the cell firing rates) but can also be used independently (for example an `Environment` and `Agent` can be modelled without any `Neurons` if users only require the motion model, or alternatively users can calculate cell activity on an imported trajectory without using the random motion model).

We envisage use cases falling into two broad categories. (i) Data generation: The user is interested in generating realistic trajectories and/or neural data for use in a downstream analysis or model training procedure (*Lee et al., 2023*). (ii) Advanced modelling: The user is interested in building a model of the brain's role in navigation (*George et al., 2023*), including how behaviour and neural representations mutually interact.

Below we briefly describe the most important details and features of RatInABox, divided into their respective classes. We leave all mathematical details to the Methods. Additional details (including example scripts and figures) can be found in the supplementary material and on the GitHub repository. The codebase itself is comprehensively documented and can be referenced for additional understanding where necessary.

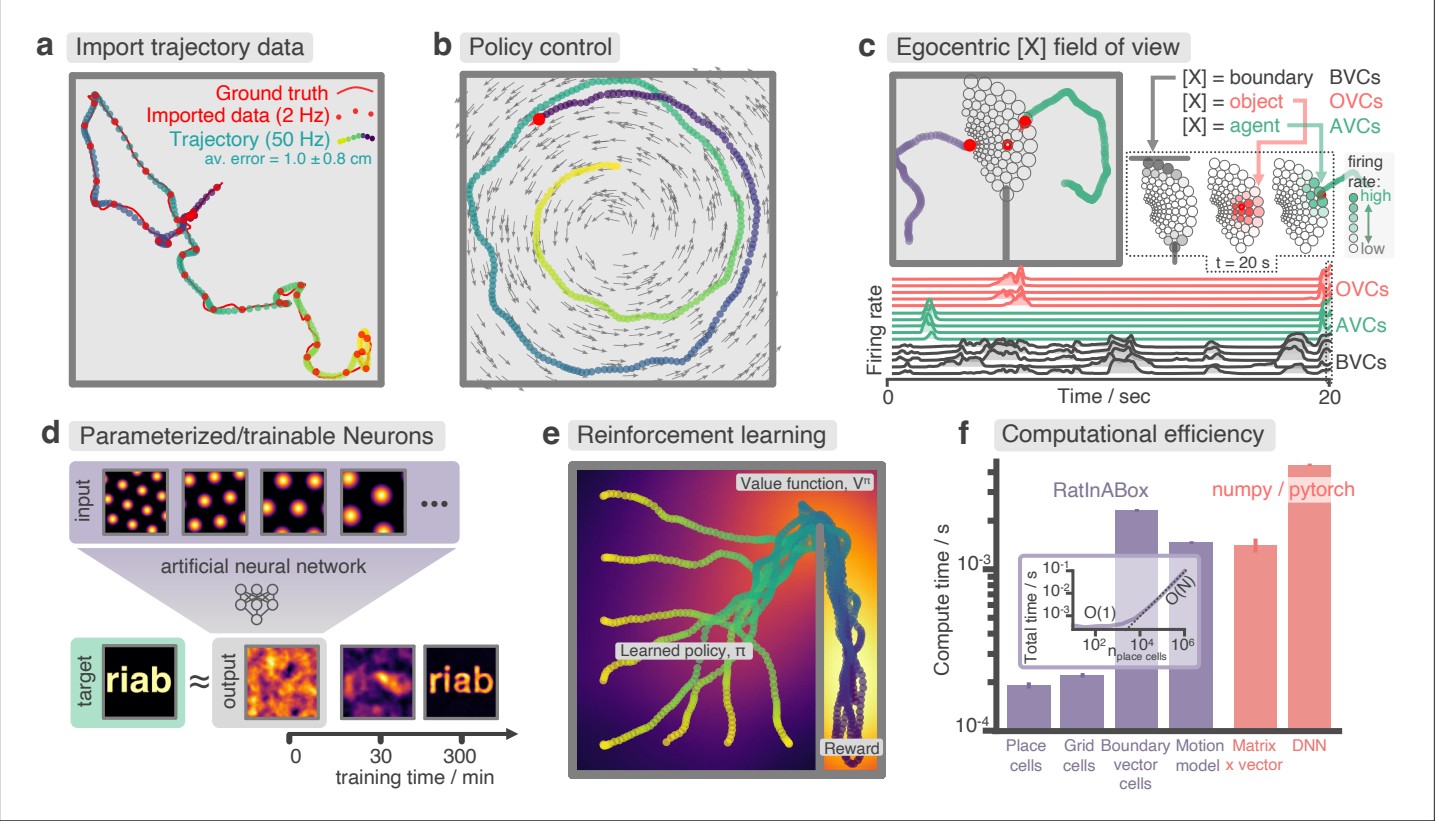

**Figure 3.** Advanced features and computational efficiency analysis. (**a**) Low temporal-resolution trajectory data (2 Hz) imported into RatInABox is upsampled ('augmented') using cubic spline interpolation. The resulting trajectory is a close match to the ground truth trajectory (*Sargolini et al., 2006*) from which the low resolution data was sampled. (**b**) Movement can be controlled by a user-provided 'drift velocity' enabling arbitrarily complex motion trajectories to be generated. Here, we demonstrate how circular motion can be achieved by setting a drift velocity (grey arrows) which is tangential to the vector from the centre of the `Environment` to the `Agent`'s position. (**c**) Egocentric `VectorCells` can be arranged to tile the `Agent`'s field of view, providing an efficient encoding of what an `Agent` can 'see'. Here, two `Agents` explore an `Environment` containing walls and an object. `Agent-1` (purple) is endowed with three populations of Boundary- (grey), Object- (red), and Agent- (green) selective field of view `VectorCells`. Each circle represents a cell, its position (in the head-centred reference frame of the `Agent`) corresponds to its angular and distance preferences and its shading denotes its current firing rate. The lower panel shows the firing rate of five example cells from each population over time. (**d**) A `Neurons` class containing a feed forward neural network learns, from data collect online over a period of 300 min, to approximate a complex target receptive field from a set of grid cell inputs. This demonstrates how learning processes can be incorporated and modelled into RatInABox. (**e**) RatInABox used in a simple reinforcement learning example. A policy iteration technique converges onto an optimal value function (heatmap) and policy (trajectories) for an Environment where a reward is hidden behind a wall. State encoding, policy control and the `Environment` are handled naturally by RatInABox. (**f**) Compute times for common RatInABox (purple) and non-RatInABox (red) operations on a consumer grade CPU. Updating the random motion model and calculating boundary vector cell firing rates is slower than place or grid cells (note log-scale) but comparable, or faster than, size-matched non-RatInABox operations. Inset shows how the total update time (random motion model and place cell update) scales with the number of place cells.

## The Environment

Unlike discretised models, where environments are stored as sets of nodes ('states') connected by edges ('actions')(*Juliani et al., 2022*), here `Environment`s are continuous domains containing walls (1D line segments through which locomotion is not allowed) and objects (which are 0-dimensional and act as visual cues). Boundaries and visual cues are thought to provide an important source of sensory data into the hippocampus (*O'Keefe and Burgess, 1996*; *Hartley et al., 2000*; *Barry et al., 2006*; *Solstad et al., 2008*) and play an important role in determining cell activity during navigation (*Stachenfeld et al., 2017*; *de Cothi and Barry, 2020*). An `Environment` can have periodic or solid boundary conditions and can be one- or two-dimensional (*Figure 1a, d*).

## The Agent

### Physically realistic random motion

Smooth and temporally continuous random motion can be difficult to model. To be smooth (and therefore physically plausible), a trajectory must be continuous in both position *and* velocity. To be temporally continuous, the statistics of the motion must be independent of the integration timestep being used. To be random, position and velocity at one time must not be reliable predictors of position and velocity at another time, provided these times are seperated by a sufficiently long interval. Implementations of random motion models typically fail to satisfy one, or sometimes two, of these principles (*Raudies and Hasselmo, 2012*; *Benna and Fusi, 2021*).

Ornstein-Uhlenbeck processes, which sit at the heart of the RatInABox random motion model, are continuous-in-time random walks with a tendency to return to a central drift value. The decorrelation timescale can be also be controlled. We use these to update the velocity vector (linear and rotational velocities are updated independently) on each update step. Position is then updated by taking a step along the velocity vector with some additional considerations to avoid walls. This method ensures both position and velocity are continuous, yet evolve 'randomly' (*Figure 1a, d*), and the statistics of the motion is independent of the size of the discretisation timestep being used.

Reanalysing rat locomotion data from *Sargolini et al., 2006* (as has been done before, by *Raudies and Hasselmo, 2012*) we found that the histograms of linear speeds are well fit by a Rayleigh distributions whereas rotational velocities are approximately fit by normal distributions (*Figure 2a*). Unlike *Raudies and Hasselmo, 2012*, we also extract the decorrelation timescale of these variables and observe that rotational velocity in real locomotion data decorrelates nearly an order of magnitude faster than linear velocity (0.08 s vs. 0.7 s). We set the default parameters of our Ornstein-Uhlenbeck processes (including applying a transform on the linear velocity so its long-run distribution also follows a Rayleigh distribution, see Methods) to those measured from the *Sargolini et al., 2006* dataset (*Figure 2b*).

### Motion near walls

Animals rarely charge head-first into a wall, turn around, then continue in the opposite direction. Instead, they slow down smoothly and turn to avoid a collision. Additionally, during random foraging, rodents are observed to show a bias towards following walls, a behaviour known as thigmotaxis (*Satoh et al., 2011*; *Figure 2c*). To replicate these observations, walls in the `Environment` lightly repel the `Agent` when it is close. Coupled with the finite turning speed this creates (somewhat counterintuitively) a thigmotactic effect where the agent over-explores walls and corners, matching what is observed in the data (*Figure 2e*). A user-defined parameter called '`thigmotaxis`' can be used to control the strength of this emergent effect (*Figure 2d*).

### Imported trajectories

RatInABox supports importing trajectory data which can be used instead of the inbuilt random motion model. Imported trajectory data points which may be of low temporal-resolution are interpolated using cubic splines and smoothly upsampled to user-define temporal precision (*Figure 3a*). This upsampling is essential if one wishes to use low temporal resolution trajectory data to generate high temporal resolution neural data.

### Trajectory control

RatInABox supports online velocity control. At each integration step a target drift velocity can be specified, towards which the `Agent` accelerates. We anticipate this feature being used to generate complex stereotyped trajectories or to model processes underpinning complex spatial behaviour (as we demonstrate in *Figure 3b, e*).

## Neurons

RatInABox provides multiple premade `Neurons` subclasses chosen to replicate the most popular and influential cell models and state representations across computational neuroscience and machine learning. A selection of these are shown in *Figure 1b*. See Methods for mathematical details. These currently include:

- `PlaceCells`: A set of locations is sampled uniformly at random from across the `Environment` or provided manually, each defining the centre of a place field. The place cell firing rate is determined by the some function of the distance from the `Agent` to the centre of the place field. Provided functions are
  - Gaussian: A Gaussian centred on the place field centre.
  - Gaussian threshold: A gaussian cropped and levelled at 1 standard deviation.
  - Difference of two Gaussians: A wide Gaussian substracted from a narrower Gaussian with zero total volume.
  - Top hat: Fires uniformly only within a circle of specific radius (similar to tile coding in machine learning).
  - One hot: Only the closest place cell to a given position will fire. This is useful for replicating tabular state spaces but with continuous motion.
  - `PhasePrecessingPlaceCells`: A subclass of `PlaceCells` which display phase precession (*O'Keefe and Recce, 1993*) with respect to a background LFP theta-oscillation.
- `GridCells`: Grid cells are modelled using a method proposed by *Burgess et al., 2007*. Receptive fields are given by the thresholded or shifted sum of three cosine waves at 60°.
- `VectorCells`: Each vector cells responds to salient features in the `Environment` at a preferred distance and angle according to a model inspired by the double-Gaussian model used by *Hartley et al., 2000*. Vector cells can be 'allocentric' (angular preferences are relative to true-North) or 'egocentric' (*Byrne et al., 2007*) (angular preferences are relative to the `Agent`'s heading). Types include:
  - `BoundaryVectorCells`: Respond to walls.
  - `ObjectVectorCells`: Respond to objects.
  - `AgentVectorCells`: Respond to other `Agent`s.
  - `FieldOfViewBVCs/OVCs/AVCs`: Egocentric vector cells arranged to tile the `Agent`'s field-of-view, further described below.
- `HeadDirectionCells`: Each cell has a preferred direction. The firing rate is given by a von Mises distribution centred on the preferred direction.
- `VelocityCells`: Like `HeadDirectionCells` but firing rate scales proportional to speed.
- `SpeedCell`: A single cell fires proportional to the scalar speed of the `Agent`.
- `RandomSpatialNeurons`: Each cell has a locally smooth but random spatial receptive field of user-defined lengthscale.

A dedicated space containing additional cell classes not described here, is made available for community contributions to this list.

## Customizable and trainable neurons

Any single toolkit cannot contain all possible neural representations of interest. Besides, static cell types (e.g. `PlaceCells`, `GridCells` etc.) which have fixed receptive fields are limiting if the goal is to study how representations and/or behaviour are learned. RatInABox provides two solutions: Firstly, being open-source, users can write and contribute their own bespoke `Neurons` (instructions and examples are provided) with arbitrarily complicated rate functions. Secondly, two types of function-approximator `Neurons` are provided which map inputs (the firing rate of other `Neurons`) to outputs (their own firing rate) through a parameterised function which can be hand-tuned or trained to represent an endless variety of receptive field functions including those which are mixed selective, non-linear, dynamic, and non-stationary.

- `FeedForwardLayer`: Calculates a weighted linear combination of the input `Neurons` with optional bias and non-linear activation function.
- `NeuralNetworkNeurons`: Inputs are passed through a user-provided artificial neural network.

Naturally, function-approximator `Neurons` can be used to model how neural populations in the brain communicate, how neural representations are learned or, in certain cases, neural dynamics. In an online demo, we show how `GridCells` and `HeadDirectionCells` can be easily combined using a `FeedForwardLayer` to create head-direction selective grid cells (aka. conjunctive grid cells *Sargolini et al., 2006*). In *Figure 3d* and associated demo `GridCells` provide input to a `NeuralNetworkNeurons` class which is then trained, on data generated during exploration, to have a highly complex and non-linear receptive field. Function-approximator `Neurons` can themselves be used as inputs to other function-approximator `Neurons` allowing multi-layer and/or recurrent networks to be constructed and studied.

## Field of view encodings

Efficiently encoding what an `Agent` can 'see' in its local vicinity, aka. its field of view, is crucial for many modelling studies. A common approach is to use a convolutional neural network (CNN) to process a rendered image of the nearby environment and extract activations from the final layer. However, this method is computationally expensive and necessitates training the CNN on a large dataset of visual images.

RatInABox offers a more efficient alternative through the use of `VectorCells`. Three variants – `FieldOfViewBVCs`, `FieldOfViewOVCs`, and `FieldOfViewAVCs` – comprise populations of *egocentric* `Boundary`-, `Object`-, and `AgentVectorCells` with angular and distance preferences specifically set to tile the `Agent`'s field of view. Being egocentric means that the cells remained fixed in the reference frame of the `Agent` as it navigates the `Environment`. Users define the range and resolution of this field of view. Plotting functions for visualising the field of view cells, as shown in *Figure 3c*, are provided.

## Geometry and boundary conditions

In RatInABox, `PlaceCells` and `VectorCells` are sensitive to walls in the `Environment`. Three distance geometries are supported: 'euclidean' geometry calculates the Euclidean distance to a place field centre and so cell activity will 'bleed' through boundaries as if they weren't there. 'line_of_sight' geometry allows a place cell to fire only if there is direct line-of-sight to the place field centre from the current location. Finally 'geodesic' geometry (default) calculates distance according to the shortest boundary-avoiding path to the cell centre (notice smooth wrapping of the third place field around the wall in *Figure 1b*). The latter two geometries respect the observation that place fields don't typical pass through walls, an observation which is thought to support efficient generalisation in spatial reinforcement learning (*Gustafson and Daw, 2011*). Boundary conditions can be periodic or solid. In the former case, place fields near the boundaries of the environment will wrap around.

## Rate maps

RatInABox simplifies the calculation and visualization of rate maps through built-in protocols and plotting functions. Rate maps can be derived explicitly from their known analytic firing functions or implicitly from simulation data. The explicit method computes rate maps by querying neuron firing rates at all positions simultaneously, utilizing 'array programming' to rapidly compute the rate map. In the implicit approach, rate maps are created by plotting a smoothed histogram of positions visited by the `Agent`, weighted by observed firing rates (a continuous equivalent of a smoothed spike raster plot). Additionally, the tool offers the option to visualize spikes through raster plots.

# Results

The default parameters of the random motion model in RatInABox are matched to observed statistics of rodent locomotion, extracted by reanalysing data from *Sargolini et al., 2006* (data freely available at: https://doi.org/10.11582/2017.00019, exact filename used: 8F6BE356-3277-475C-87B1-C7A977632DA7_1/11084–03020501_t2c1.mat). Trajectories and statistics from the real data (*Figure 2a*) closely compare to the artificially generated trajectories from RatInABox (*Figure 2b*). Further, data (*Satoh et al., 2011*) shows that rodents have a tendency to over-explore walls and corners, a bias often called 'thigmotaxis' which is particularly pronounced when the animal is new to the environment (*Figure 2c*). This bias is correctly replicated in the artificial trajectories generated by RatInABox - the strength of which can be controlled by a single parameter `Agent.thigmotaxis` (*Figure 2d, e*).

RatInABox can import and smoothly interpolate user-provided trajectory data. This is demonstrated in *Figure 3a* where a low-resolution trajectory is imported into RatInABox and smoothly upsampled using cubic spline interpolation. The resulting trajectory is a close match to the ground truth. Note that without upsampling, this data (2 Hz) would be far too low in temporal-resolution to usefully simulate neural activity. For convenience, the exact datafile *Sargolini et al., 2006* used in *Figures 3a and 2a* is uploaded with permission to the GitHub repository and can be imported using `Agent.import_trajectory(dataset="sargolini")`. An additional trajectory dataset from a much larger environment is also supplied with permission from *Tanni et al., 2022*.

RatInABox is computationally efficient. We compare compute times for typical RatInABox operations (**Figure 3f**, purple bars) to typical *non*-RatInABox operations representing potential 'bottlenecking' operations in a downstream analysis or model-training procedure for which RatInABox is providing data (**Figure 3f**, red bars). These were multiplying a matrix by a vector using the numpy (**Harris et al., 2020**) package and a forward and backward pass through a small feedforward artificial neural network using the pytorch package (**Paszke et al., 2019**). `PlaceCells`, `GridCells` and the random motion model all update faster than these two operations. `BoundaryVectorCells` (because they require integrating around a 360° field-of-view) are significantly slower than the other cells but still outpace the feedforward neural network. All vector, matrix, and cell populations were size $n = 100$, the feed forward network had layer sizes $n_L = (100, 1000, 1000, 1)$, the `Environment` was 2D with no additional walls and all operations were calculated on a consumer-grade CPU (MacBook Pro, Apple M1). These results imply that, depending on the details of the use-case, RatInABox will likely not be a significant computational bottleneck.

Our testing (**Figure 3f**, inset) reveals that the combined time for updating the motion model and a population of `PlaceCells` scales sublinearly O(1) for small populations $n > 1000$ where updating the random motion model dominates compute time, and linearly for large populations $n < 1000$. `PlaceCells`, `BoundaryVectorCells` and the `Agent` motion model update times will be additionally affected by the number of walls/barriers in the `Environment`. 1D simulations are significantly quicker than 2D simulations due to the reduced computational load of the 1D geometry.

## Case studies

We envisage RatInABox being used to support a range of theoretical studies by providing data and, if necessary, infrastructure for building models powered by this data. This 'Bring-Your-Own-Algorithm' approach makes the toolkit generally applicable, not specialised to one specific field. Two examplar use-cases are provided in the supplement and are briefly described below. The intention is to demonstrate the capacity of RatInABox for use in varied types of computational studies and to provide tutorials as a tool for learning how to use the package. Many more demonstrations and accompanying notebooks are provide on the Github repository.

In our first example, we perform a simple experiment where location is decoded from neural firing rates (**Appendix 1—figure 1**). Data – the location and firing rate trajectories of an `Agent` randomly exploring a 2D `Environment` – are generated using RatInABox. Non-parametric Gaussian process regression is used to predict position from firing rates on a held-out testing dataset. We compare the accuracy of decoding using different cell types; place cells, grid cells and boundary vector cells.

Next, we demonstrate the application of RatInABox to a simple reinforcement learning (RL) task (**Appendix 1—figure 2**, summarised in **Figure 3e**). A small network capable of model-free RL is constructed and trained using RatInABox. First a neuron calculates and learns – using a continuous variant of temporal difference learning – the value function $V^{\pi}(x) = \sum_i w_i F_i^{\mathrm{pc}}(x)$ as a linear combination

of place cell basis features. Then a new 'improved' policy is defined by setting a drift velocity – which biases the `Agent`'s motion – proportional to the gradient of the value function $v^{\mathrm{drift}}(x) = \pi(x) \propto \nabla_x V^{\pi}$. The `Agent` is therefore encouraged to move towards regions with high value. Iterating between these stages over many episodes ('policy iteration') results in convergence towards near optimal behaviour where the `Agent` takes the shortest route to the reward, avoiding the wall (**Figure 3e**).

Additional tutorials, not described here but available online, demonstrate how RatInABox can be used to model splitter cells, conjunctive grid cells, biologically plausible path integration, successor features, deep actor-critic RL, whisker cells and more. Despite including these examples we stress that they are not exhaustive. RatInABox provides the framework and primitive classes/functions from which highly advanced simulations such as these can be built.

## Discussion

RatInABox is a lightweight, open-source toolkit for generating realistic, standardised trajectory and neural data in continuous environments. It should be particularly useful to those studying spatial navigation and the role of the hippocampal formation. It remains purposefully small in scope - intended primarily as a means for generating data. We do not provide, nor intend to provide, a set of benchmark learning algorithms to use on the data it generates. Its user-friendly API, inbuilt data-plotting

functions and general yet modular feature set mean it is well placed to empower a wide variety of users to more rapidly build, train and validate models of hippocampal function (*Lee et al., 2023*) and spatial navigation (*George et al., 2023*), accelerating progress in the field.

Our package is not the first to model neural data (*Stimberg et al., 2019*; *Hepburn et al., 2012*; *Hines and Carnevale, 1997*) or spatial behaviour (*Todorov et al., 2012*; *Merel et al., 2019*), yet it distinguishes itself by integrating these two aspects within a unified, lightweight framework. The modelling approach employed by RatInABox involves certain assumptions:

1. It does not engage in the detailed exploration of biophysical (*Stimberg et al., 2019*; *Hines and Carnevale, 1997*) or biochemical (*Hepburn et al., 2012*) aspects of neural modelling, nor does it delve into the mechanical intricacies of joint and muscle modelling (*Todorov et al., 2012*; *Merel et al., 2019*). While these elements are crucial in specific scenarios, they demand substantial computational resources and become less pertinent in studies focused on higher-level questions about behaviour and neural representations.
2. A focus of our package is modelling experimental paradigms commonly used to study spatially modulated neural activity and behaviour in rodents. Consequently, environments are currently restricted to being two-dimensional and planar, precluding the exploration of three-dimensional settings. However, in principle, these limitations can be relaxed in the future.
3. RatInABox avoids the oversimplifications commonly found in discrete modelling, predominant in reinforcement learning (*Maxime et al., 2023*; *Juliani et al., 2022*) which we believe impede its relevance to neuroscience.
4. Currently, inputs from different sensory modalities, such as vision or olfaction, are not explicitly considered. Instead, sensory input is represented implicitly through efficient allocentric or egocentric representations. If necessary, one could use the RatInABox API in conjunction with a third-party computer graphics engine to circumvent this limitation.
5. Finally, focus has been given to generating synthetic data from steady-state systems. Hence, by default, `Agent`s and `Neuron`s do not explicitly include learning, plasticity or adaptation. Nevertheless we have shown that a minimal set of features such as parameterised function-approximator neurons and policy control enable time varying behavioural policies and cell responses (*Bostock et al., 1991*; *Barry et al., 2007*) to be modelled within the framework.

In conclusion, while no single approach can be deemed the best, we believe that RatInABox's unique positioning makes it highly suitable for normative modelling and NeuroAI. We anticipate that it will complement existing toolkits and represent a significant contribution to the computational neuroscience toolbox.

## Materials and methods

The following section describes in mathematical detail the models used within RatInABox. Table 1, below compiles a list of all important parameters along with their default values, allowed ranges and how they can be adjusted. These are up to date as of the time/version of publication but later versions may differ, see the GitHub repository for the most up-to-date list.

### Motion model
#### Temporally continuous random motion
Our random motion model is based on the Ornstein Uhlenbeck (OU) process, $X_{\theta,\lambda,\mu}(t)$, a stochastic process satisfying the Langevin differential equation

$$X_{\theta,\lambda,\mu}(t + dt) = X_{\theta,\lambda,\mu}(t) + dX_{\theta,\lambda,\mu}(t),$$
$$dX_{\theta,\lambda,\mu}(t) = \theta(\mu - X_{\theta,\lambda,\mu}(t))dt + \lambda\eta(t)\sqrt{dt} \tag{1}$$

where $\eta(t) \sim \mathcal{N}(0, 1)$ is Gaussian white noise and $\theta$, $\lambda$ and $\mu$ are constants. The first term in the update equation drives decay of $X_{\theta,\lambda,\mu}(t)$ towards the mean $\mu$. The second term is a stochastic forcing term, driving randomness. These stochastic processes are well studied; their unconditioned covariance across time is

$$\langle X_{\theta,\lambda,\mu}(t)X_{\theta,\lambda,\mu}(t') \rangle = \frac{\lambda^2}{2\theta}e^{-\theta|t-t'|}. \tag{2}$$

Thus $X_{\theta,\lambda,\mu}(t)$ decorrelates smoothly over a timescale of $\tau = 1/\theta$. Over long periods $X_{\theta,\lambda,\mu}(t)$ is stochastic and therefore unpredictable. Its long-run stationary probability distribution is a Gaussian with mean $\mu$ and standard deviation $\sigma = \sqrt{\lambda^2/2\theta}$. We can re-parameterise the Ornstein Uhlenbeck process in terms of these more intuitive parameters (the decoherence timescale $\tau$ and the long-run standard deviation $\sigma$) using the transformations

$$\theta = \frac{1}{\tau} \quad , \quad \lambda = \sqrt{\frac{2\sigma^2}{\tau}}, \tag{3}$$

to give

$$X_{\tau,\sigma,\mu}(t + dt) = X_{\tau,\sigma,\mu}(t) + dX_{\tau,\sigma,\mu}(t),$$

$$dX_{\tau,\sigma,\mu}(t) = \frac{1}{\tau}(\mu - X_{\tau,\sigma,\mu}(t))dt + \sqrt{\frac{2\sigma^2}{\tau}}\eta(t)\sqrt{dt}. \tag{4}$$

Ornstein Uhlenbeck processes have the appealing property that they are temporally continuous (their statistics are independent of $dt$) and allow for easy control of the long-run standard deviation and the decoherence timescale of the stochastic variable. For these reasons, we use use them to model rotational and linear velocities within RatInABox.

## 2D motion

For 2D locomotion, we sample the `Agent`'s rotational velocity $\omega(t) = \dot{\theta}_v(t)$ and linear speed, $v_{2D}(t) = \|\mathbf{v}(t)\|$, from independent OU processes. This is because, as shown in the Results section, they have decoherence timescales differing by an order of magnitude. Rotational velocity is sampled from a standard Ornstein Uhlenbeck process with zero mean. Linear speed is also sampled from an Ornstein Uhlenbeck process with one additional transform applied in order to match the observation that linear speeds have a Rayleigh, not normal, distribution.

$$\omega(t) \sim X_{\tau_\omega,\alpha_\omega,0}(t), \tag{5}$$

$$v_{2D}(t) = R_{\sigma_v}(z(t)) \quad \text{where} \quad z(t) \sim X_{\tau_v,1,0}(t), \tag{6}$$

where $R_\sigma(x)$ is a monotonic transformation which maps a normally distributed random variable $x \sim \mathcal{N}(0,1)$ to one with a Rayleigh distribution of scale parameter $\sigma$ corresponds to the mode, or $\approx 0.8$ times the mean, of the Rayleigh distribution.

$$R_\sigma(x) = \sigma\sqrt{-2\ln\left(1 - \frac{1}{2}\left[1 + \text{erf}\left(\frac{x}{\sqrt{2}}\right)\right]\right)}. \tag{7}$$

The parameters $\{\tau_\omega, \sigma_\omega, \tau_v, \sigma_v\}$ are fitted from real open field 2D locomotion data in **Figure 2** or can be set by the user (see Table 1, below).

Full trajectories are then sampled as follows: First the rotational and linear velocities are updated according to **Equations 5, 6** (and additional considerations for walls, see next section). Next the velocity direction, $\theta_v(t)$ – defined as the angle of the velocity vector measured anticlockwise from the x-direction – is updated according to the rotational velocity, $\omega(t)$.

$$\theta_v(t) = \left(\theta_v(t - dt) + \omega(t)dt\right) \bmod 2\pi. \tag{8}$$

This is combined with the linear speed, $v_{2D}(t)$ to calculate new total velocity vector, $\mathbf{v}(t)$.

$$\mathbf{v}(t) = v_{2D}(t)\begin{bmatrix} \cos\theta_v(t) \\ \sin\theta_v(t) \end{bmatrix}. \tag{9}$$

Finally position, $\mathbf{x}(t)$, is updated by integrating along the total velocity vector to give a continuous and smooth, but over long time periods random, motion trajectory.

$$\mathbf{x}(t) = \mathbf{x}(t - dt) + \mathbf{v}(t)dt. \tag{10}$$

## 1D motion

Motion in 1D is more simple than motion in 2D. Velocity is also modelled as an Ornstein Uhlenbeck process without the Rayleigh transform. In this case a non-zero mean, $\mu_v$, corresponding to directional bias in the motion, can be provided by the user. In summary:

$$v_{1D}(t) \sim X_{\tau_v, \alpha_v, \mu_v}(t), \tag{11}$$

$$x(t) = x(t - dt) + v_{1D}(t)dt. \tag{12}$$

## External velocity control

It is possible to provide an external velocity signal controlling the `Agent`'s motion. After the random motion update (as described above) is applied, if an external velocity $\mathbf{v}_{drift}(t)$ is provided by the user, an additional update to the velocity vector is performed

$$d\mathbf{v}(t + dt) = \frac{1}{\tau_{drift}}(\mathbf{v}_{drift}(t) - \mathbf{v}(t))dt. \tag{13}$$

In cases where $\tau_{drift} >> \tau_v$ the net update to the velocity (random update and drift update) is dominated by the random component. When $\tau_{drift} << \tau_v$ the update is dominated by the drift component. We define $\tau_{drift} := \tau_v/k$ where $k$ is an argument also provided by the user. To good approximation for large $k >> 1$ the `Agent` velocity closely tracks the drift velocity at all times and is not random whilst for $k << 1$ the drift velocity is ignored and the motion is entirely random.

## Motion near walls in 2D

An important feature is the ability to generate `Environment`s with arbitrary arrangements of walls (aka 'barriers' or 'boundaries'). Walls are meaningful only if they appropriately constrain the motion of the `Agent`. For biological agents this means three things:

1. The `Agent` cannot travel through a wall.
2. The `Agent` slows down upon approaching a wall to avoid a full-speed collision.
3. There may be a bias called "thigmotaxis" for the `Agent` to stay near walls.

Our motion model replicates these three effects as follows:

### Collision detection

To avoid travelling through walls, if a collision is detected the velocity is elastically reflected off the wall (normal component is flipped). The speed is then scaled to one half the average motion speed, $v_{2D}(t) = 0.5\sigma_v$.

### Wall repulsion

**Spring-deceleration model.** In order to slow down *before* colliding with a wall the `Agent` feels an acceleration, perpendicular to the wall, whenever it is within a small distance, $d_{wall}$, of the wall.

$$\dot{\mathbf{v}}(t) = k_1 \sum_{walls, j} \mathbf{n}_j \begin{cases} \frac{(s \cdot \sigma_v)^2}{d_{wall}^2} \cdot (d_{wall} - d_{\perp, j}(t)) & \text{if } d_{\perp, j}(t) \leq d_{wall}, \\ 0 & \text{if } d_{\perp, j}(t) > d_{wall}. \end{cases} \tag{14}$$

$d_{\perp, j}(t)$ is the perpendicular distance from the `Agent` to the $j^{th}$ wall, $\mathbf{n}_j$ is the perpendicular norm of the $j^{th}$ wall (the norm pointing towards the `Agent`) and $k_1$ & $s$ are constants (explained later). $d_{wall}$ is the distance from the wall at which the `Agent` starts to feel the deceleration, defaulting to $d_{wall} = 0.1$ m.

Note that this acceleration is identical to that of an oscillating spring-mass where the base of the spring is attached a distance $d_{wall}$ from the wall on a perpendicular passing through the `Agent`. The spring constant is tuned such that a mass starting with initial velocity towards the wall of $-s\sigma_v \mathbf{n}_j$ would stop *just* before the wall. In summary, for $k_1 = 1$, if the `Agent` approaches the wall head-on at speed of $s\sigma_v$ ($s$ times its mean speed) this deceleration will just be enough to avoid a collision.

$s$ is the unitless wall repel strength parameter (default $s = 1$). When it is high, walls repel the agent strongly (only fast initial speeds will result in the agent reaching the wall) and when it is low, walls repel

weakly (even very slow initial speeds will not be slowed done by the spring dynamics). When $s = 0$ wall repulsion is turned off entirely.

**Conveyor-belt model.** A second (similar, but not exactly equivalent) way to slow down motion near a wall is to consider a hypothetical conveyor belt near the wall. This conveyor belt has a non-uniform velocity pointing away from the wall of

$$\dot{\mathbf{x}}(t) = k_2 \sum_{\text{walls},j} \mathbf{n}_j \begin{cases} s \cdot \sigma_{\text{v}} \left( 1 - \sqrt{1 - \dfrac{(d_{\text{wall}} - d_{\perp,j}(t))^2}{d_{\text{wall}}^2}} \right) & \text{if } d_{\perp,j}(t) \leq d_{\text{wall}}, \\ 0 & \text{if } d_{\perp,j}(t) > d_{\text{wall}}. \end{cases} \tag{15}$$

When the `Agent` is close to the wall the hypothetical conveyor-belt moves it backwards on each time step, effectively slowing it down. Note that this velocity is identical to that of a spring-mass attached to the wall with initial velocity $s\sigma_{\text{v}}\mathbf{n}_j$ away from the wall and spring constant tuned to stop the mass just before it reaches a distance $d_{\text{wall}}$. In summary, for $k_2 = 1$, if the `Agent` approaches the wall head-on at speed of $s\sigma_{\text{v}}$ the conveyor belt will just be fast enough to bring it to a halt at the location of the wall.

**Wall attraction (thigmotaxis).** Although similar, there is an exploitable difference between the 'spring-deceleration' and 'conveyor-belt' models: the 'conveyor-belt' changes the `Agent`s position, $\mathbf{x}(t)$, on each step but not its internal velocity variable $\mathbf{v}(t)$. As as result (and as the conveyor-belt intuition suggests) it will slow down the `Agent`'s approach towards the wall without causing it to turn around. This creates a 'lingering' or 'thigmotactic' effect whereby whenever the `Agent` heads towards a wall it may carry on doing so, without collision, for some time until the stochastic processes governing its motion (section 'Temporally continuous random motion') cause it to turn. Conversely the 'spring-deceleration' model has no 'thigmotactic' effect since it actively changes the internal velocity variable causing the `Agent` to turn around or 'bounce' off the walls.

The relative strengths of these two effects, $k_1$ and $k_2$, are controlled by a single thigmotaxis parameter, $\lambda_{\text{thig}} \in [0, 1]$ which governs the trade-off between these two models.

$$k_1 = 3(1 - \lambda_{\text{thig}})^2, \qquad k_2 = 6\lambda_{\text{thig}}^2. \tag{16}$$

When $\lambda_{\text{thig}} = 1$ only the conveyor belt model is active giving a strong thigmotactic effects. When $\lambda_{\text{thig}} = 0$ only the spring-deceleration model is active giving no thigmotactic effect. By default $\lambda_{\text{thig}} = 0.5$. The constants 3 and 6 are tuning parameters chosen by hand in order that direct collisions with the walls are rare but not impossible.

Although this procedure, intended to smoothly slow the `Agent` near a wall, may seem complex, it has a two advantages: Firstly, deceleration near walls is smooth, becoming stronger as the `Agent` gets nearer and so induces no physically implausible discontinuities in the velocity. Secondly, it provides a tunable way by which to control the amount of thigmotaxis (evidenced in *Figure 2c, d*). Recall that these equations only apply to motion very near the wall ($< d_{\text{wall}}$) and they can be turned off entirely ($s = 0$) (see Table 1, below).

## Importing trajectories

Users can override the random motion model by importing their own trajectory with `Agent.import_trajectory(times,positions)` where times is an array of times (not necessarily evenly spaced) and positions is an array of positions at each time. The trajectory is then interpolated using scipy.interpolate's interp1d function following which the standard RatInABox `Agent.update(dt)` API is called to move the `Agent` to a new position a time dt along the imported trajectory.

When moving along imported trajectories the `Agent` will not be subject to the wall repel nor wall collision effects described above.

## Head direction

As well as position and velocity `Agent`s have a head direction, $\hat{\mathbf{h}}(t)$. Head direction is used by various cell types to determine firing rate including `HeadDirectionCells` and (egocentric) `VectorCells`.

By default, head direction is just the smoothed-then-normalised velocity vector, updated on each timestep as follows:

$$\mathbf{h}(t + dt) = \left( 1 - \frac{dt}{\tau_h} \right) \hat{\mathbf{h}}(t) + \frac{dt}{\tau_h} \frac{\mathbf{v}(t)}{\|\mathbf{v}(t)\|} \tag{17}$$

$$\hat{\mathbf{h}}(t + dt) = \frac{\mathbf{h}(t + dt)}{\|\mathbf{h}(t + dt)\|}. \tag{18}$$

By default the amount of smoothing is very small (in 2D $\tau_h = 0.15$, in 1D there is no smoothing at all) meaning that, to a good approximation, head direction is simply the normalised velocity vector at time $t$, $\hat{\mathbf{h}}(t) \approx \hat{\mathbf{v}}(t)$. However by storing head direction as an independent variable, we make available the possibility for users to craft their own, potentially more complex, head direction dynamics if desired.

We also define the head direction angle $\phi_h(t)$ aka. the angle of head direction vector measured clockwise from the x-axis.

## Distance measures

In many of the cell models, it is necessary to calculate the 'distance' between two locations in the `Environment` (for example to calculate the firing rate of a Gaussian `PlaceCell`). This might depend on the type of geometry being used and the arrangement of walls in the `Environment`. There are three types of geometry currently supported:

$$\texttt{euclidean} : d(\mathbf{x}_1, \mathbf{x}_2) = \|\mathbf{x}_1 - \mathbf{x}_2\| \tag{19}$$

$$\texttt{geodesic} : d(\mathbf{x}_1, \mathbf{x}_2) = \text{length of shortest wall-avoiding path between } \mathbf{x}_1 \text{ and } \mathbf{x}_2 \tag{20}$$

$$\texttt{line\_of\_sight}: d(\mathbf{x}_1, \mathbf{x}_2) = \begin{cases} \|\mathbf{x}_1 - \mathbf{x}_2\|, & \text{if no wall obstructs the straight line between } \mathbf{x}_1 \text{ and } \mathbf{x}_2 \\ \infty, & \text{otherwise} \end{cases} \tag{21}$$

By default RatInABox typically uses geodesic distance, except in `Environments` with more than one additional wall where calculating the shortest path becomes computationally expensive. In these cases, `line_of_sight` distance is typically used instead. Furthermore, in `Environments` with periodic boundary conditions these distance measures will respect the periodicity by always using the shortest path between two points, wrapping around boundaries if necessary. These geometry considerations are what allow RatInABox cell classes to interact sensibly with walls (e.g. by default place cells won't bleed through walls, as observed in the brain). Hereon we refer to this as the 'environmental-distance'.

## Cell models

In the following section, we list mathematical models for some of the default provided `Neurons` subclasses, including all those covered in this manuscript. More cell types and documentation can be found on the codebase. Readers will note that, oftentimes, parameters are set randomly at the point of initialisation (e.g. where the place cells are located, the orientation of grid cells, the angular preference of boundary vector cells etc.). Many of these random parameters are all set as class attributes and so can be redefined after initialisation if necessary. For simplicity here we describe default behaviour only – the default values for all parameters and how to change them are given in Table 1, below.

**Maximum and minimum firing rates.** For most cell classes it is also possible to set their maximum and minimum firing rates ($f_{\max}$, $f_{\min}$). For simplicity, the formulae provided below are written such that they have a maximum firing rate of 1.0 Hz and minimum firing rate of 0.0 Hz but readers should be aware that after evaluation these firing rates are linearly scaled according to

$$F(t) \leftarrow (f_{\max} - f_{\min})F(t) + f_{\min}. \tag{22}$$

**Noise.** By default all `Neurons` are noiseless with their firing rates entirely determined by the deterministic mathematical models given below. Smooth Ornstein Uhlenbeck sampled random noise of coherence timescale $\tau_\eta$ and magnitude $\sigma_\eta$ can be added:

$$\eta(t) \sim \mathsf{X}_{\tau_\eta, \sigma_\eta, 0}(t) \tag{23}$$

$$F(t) \leftarrow F(t) + \eta(t) \tag{24}$$

**Rates vs. Spikes.** RatInABox `Neurons` are fundamentally rate-based. This means that their firing rate is a continuous function of time. Simultaneously, at every time-step, spikes are sampled from this firing rate and saved into the history dataframe in case spiking data is required:

$$\text{P}(\texttt{Neuron}\ i\ \text{spikes in}[t, t + dt]) = F_i(t)dt. \tag{25}$$

## PlaceCells

A set of locations (the centre of the place fields), $\{\mathbf{x}_i^{\text{PC}}\}$, is randomly sampled from the `Environment`. By default these locations sit on a grid uniformly spanning the `Environment` to which a small amount of random jitter, half the scale of the sampled grid, is added. Thus, place cell locations appear 'random' but initialising in this way ensures all parts of the `Environment` are approximately evenly covered with the same density of place fields.

The environmental-distance from the `Agent` to the place field centres is calculated ($d_i(t) = d(\mathbf{x}_i^{\text{PC}}, \mathbf{x}(t))$). The firing rate is then determined by one of the following functions (defaulting to $F^{\text{gaussian}}$):

$$F_i^{\texttt{gaussian}}(t) = e^{-d_i^2/2w_i^2} \tag{26}$$

$$F_i^{\texttt{gaussian\_threshold}}(t) = \max\left(0, \frac{e^{-d_i^2/2w_i^2} - e^{-1/2}}{1 - e^{-1/2}}\right) \tag{27}$$

$$F_i^{\texttt{diff\_of\_gaussians}}(t; r = 1.5) = \frac{e^{-d_i^2/2w_i^2} - (1/r^2)e^{-d_i^2/2(rw_i)^2}}{1 - 1/r^2} \tag{28}$$

$$\text{F}_i^{\texttt{top\_hat}}(t) = \begin{cases} 1 & \text{if } d_i \leq w_i \\ 0 & \text{otherwise} \end{cases} \tag{29}$$

$$F_i^{\texttt{one\_hot}}(t) = \delta(i == \text{argmin}_j(d_j)). \tag{30}$$

Where used, $w_i$ is the user-provided radius (aka. width) of the place cells (defaulting to 0.2 m).

## GridCells

Each grid cell is assigned a random wave direction $\theta_i \sim \mathcal{U}_{[0,2\pi]}$, gridscale $\lambda_i \sim \mathcal{U}_{[0.5\,\text{m},1.0\,\text{m}]}$ and phase offset $\phi_i \sim \mathcal{U}_{[0,2\pi]}$. The firing rate of each grid cell is given by the thresholded sum of three cosines

$$F_i(t) = \frac{1}{3}\max\left(0, \cos\left(2\pi\frac{\mathbf{x}(t) \cdot \mathbf{e}_{\theta_i}}{\lambda_i} + \phi_i\right) + \cos\left(2\pi\frac{\mathbf{x}(t) \cdot \mathbf{e}_{\theta_i+\pi/3}}{\lambda_i} + \phi_i\right) + \cos\left(2\pi\frac{\mathbf{x}(t) \cdot \mathbf{e}_{\theta_i+2\pi/3}}{\lambda_i} + \phi_i\right)\right). \tag{31}$$

$\mathbf{e}_\theta$ is the unit vector pointing in the direction $\theta$. We also provide a shifted (as opposed to rectified) sum of three cosines grid cell resulting in softer grid fields

$$F_i(t) = \frac{2}{3}\left(\frac{1}{3}\left(\cos\left(2\pi\frac{\mathbf{x}(t) \cdot \mathbf{e}_{\theta_i}}{\lambda_i} + \phi_i\right) + \cos\left(2\pi\frac{\mathbf{x}(t) \cdot \mathbf{e}_{\theta_i+\pi/3}}{\lambda_i} + \phi_i\right) + \cos\left(2\pi\frac{\mathbf{x}(t) \cdot \mathbf{e}_{\theta_i+2\pi/3}}{\lambda_i} + \phi_i\right)\right) + \frac{1}{2}\right) \tag{32}$$

$\mathbf{e}_\theta$ is the unit vector pointing in the direction $\theta$.

## VectorCells (parent class only)

`VectorCells` subclasses include `BoundaryVectorCells`, `ObjectVectorCells` and `AgentVectorCells` as well as `FieldOfView` versions of these three classes. The common trait amongst all types of `VectorCell` is that each cell is responsive to a *feature* of the environment (boundary segments, objects, other agents) at a preferred distance and angle. The firing rate of each vector cell is given by the product of two functions; a Gaussian radial function and a von Mises angular function.

When the agent is a euclidean distance $d(t)$ from the feature, at an angle $\phi(t)$ the contribution of that feature to the total firing rate is given by

$$g_i(r(t), \theta(t)) = \exp\left(-\frac{(d_i - d(t))^2}{2\sigma_{d,i}^2}\right) \cdot f_{VM}(\phi(t)|\phi_i, \kappa_i) \tag{33}$$

where $f_{VM}$ is the radial von Mises distribution (a generalisation of a Gaussian for periodic variables)

$$f_{VM}(\phi(t)|\phi_i, \kappa_i) := \exp(\kappa_i \cos(\phi(t) - \phi_i)). \tag{34}$$

Total firing rate is calculated by summing/integrating these contributions over all features in the `Environment` as described in the following sections. Distance and angular tuning parameters and defined/sampled as follows:

- $d_i$ is the distance tuning of the vector cell. By default $d_i \sim \mathcal{U}_{[0.05\text{ m},0.3\text{ m}]}$
- $\sigma_{d,i}$ is the distance tuning width. By default this increases linearly as a function of $d_i$: $\sigma_{d,i} = d_i/\beta + \xi$ for constants $\beta$ and $\xi$ but can be set otherwise. See Table 1, below for the values which are chosen to match those used by Cothi and Barry (**de Cothi and Barry, 2020**).
- $\phi_i$ is the angular tuning of the vector cell. By default $\phi_i \sim \mathcal{U}_{[0°,360°]}$.
- $\sigma_{\phi,i}$ (which defines the von Mises concentration measure $\kappa_i := 1/\sqrt{\sigma_{\phi,i}}$) is the angular tuning width of the vector cell. By default $\sigma_{\phi,i} \sim \mathcal{U}_{[10°,30°]}$.

The asymptotic equivalence between a Gaussian and a von Mises distribution (true for small angular tunings whereby von Mises distributions of concentration parameter $\kappa$ approach Gaussian distributions of variance $\sigma^2 = 1/\kappa$) means this model is effectively identical to the original boundary vector cell model proposed by **Hartley et al., 2000** but with the difference that our vector cells (BVCs included) will not show discontinuities if they have wide angular tunings of order $360°$.

All vector cells can be either

- allocentric (default): $\phi(t)$ is the angle subtended between the x-direction vector $\mathbf{e}_x = [1, 0]$, and the line between the `Agent` and the feature.
- egocentric: $\phi(t)$ is the angle subtended between the heading direction of the agent $\hat{\mathbf{h}}(t)$, and the line between the `Agent` and the feature.

## BoundaryVectorCells

The environmental features which `BoundaryVectorCells` (BVCs) respond to are the boundary segments (walls) of the `Environment`. The total firing rate of of each cell is given by integrating (computationally we use a default value of $d\theta = 2°$ to numerically approximate this integral) the contributions from the nearest line-of-sight boundary segments (walls occluded by other walls are not considered) around the full $2\pi$ field-of-view;

$$F_i(t) = K_i \int_0^{2\pi} g_i(r, \theta)d\theta, \tag{35}$$

(computationally we use a default value of $d\theta = 2°$ to numerically approximate this integral). $K_i = 1/\max_{\mathbf{x}} F_i(\mathbf{x})$ is a normalisation constant calculated empirically at initialisation such that each BVC has a maximum firing rate (before scaling) of 1.0 Hz.

## ObjectVectorCells

`ObjectVectorCells` (OVCs) respond to objects in the `Environment`. Objects are zero-dimensional and can be added anywhere within the `Environment`, each object, $j$, comes with a "type" attribute, $t_j$. Each object vector cell has a tuning type, $t_i$, and is only responsive to objects of this type. The total firing rate of of each cell is given by the sum of the contributions from all objects of the correct type in the `Environment`;

$$F_i(t) = \sum_{\text{objects},j \text{ if } t_j = t_i} g_i(r_j(t), \theta_j(t)). \tag{36}$$

Since *Equation 33* has a maximum value of 1 by definition the maximum firing rate of an object vector cell is also 1 Hz (unless multiple objects are closeby) and no normalisation is required.

### AgentVectorCells

`AgentVectorCells` respond to other `Agent`s in the `Environment`. All cells in a given class are selective to the same `Agent`, index $j$. The firing rate of each cell is then given by;

$$F_i(t) = g_i(r_j(t), \theta_j(t)). \tag{37}$$

### FieldOfViewBVCs, FieldOfViewOVCs, and FieldOfViewAVCs

`FieldOfViewBVCs/OVCs/AVCs` are a special case of the above vector cells where the tuning parameters ($d_i$, $\sigma_{d,i}$, $\phi_i$, $\sigma_{\phi,i}$) for a set of `VectorCells` are carefully set so that cells tile a predefined 'field of view'. By default these cells are egocentric and so the field of view (as the name implies) is defined relative to the heading direction of the `Agent`; if the `Agent` turns the field of view turns with it.

Users define the angular and radial extent of the field of view as well as the resolution of the cells which tile it. There is some flexiblity for users to construct complex fields of view but baic API simplifies this process, exposing a few key parameters:

- $r_{\text{fov}} = [r_{\text{fov}}^{\min}, r_{\text{fov}}^{\max}]$ (default [0.02 m, 0.2 m]): the radial extent of the field of view.
- $\theta_{\text{fov}}$ (default [0°, 75°]): the angular extend of the field of view (measured from the forward heading direction, symmetric left and right).
- $\delta_{\text{fov}}^0$ (default 0.02 m): `FieldOfView VectorCells` all have approximately circular receptive fields (i.e. the radial Gaussian and angular von Mises in *Equation 33* have matched variances which depend on their tuning distance; $\sigma_{d,i} = d_i \cdot \sigma_{\phi,i} := \delta_{\text{fov}}(d_i)$). $\delta_{\text{fov}}^0$ sets the resolution of the inner-most row of cells in the field of view, $\delta_{\text{fov}}^0 = \delta_{\text{fov}}(d_i = r_{\text{fov}}^{\min})$.
- Manifold type: For "diverging" manifolds (default) cells further away from the `Agent` have larger receptive fields $\delta_{\text{fov}}(d_i) = \xi_0 + d_i/\beta$ for user-defined $\beta$ (default $\beta = 5$) and $\xi_0 := \delta_{\text{fov}}^0 - r_{\text{fov}}^{\min}/\beta$. For "uniform" manifold all cells have the same sized receptive fields, $\delta_{\text{fov}}(d_i) = \delta_{\text{fov}}^0$.

More complex field of views can be constructed and a tutorial is provided to show how.

### HeadDirectionCells

In 2D `Environment`s each head direction cell has an angular tuning mean $\theta_i$ and width $\sigma_i := 1/\sqrt{\kappa_i}$. The response function is then a von Mises in the head direction of the `Agent`:

$$F_i(t) = \exp(\kappa_i \cos(\theta_h(t) - \theta_i)). \tag{38}$$

By default all cells have the same angular tuning width of 3° and tuning means even spaced from 0° to 360°.

In 1D `Environment`s there is always and only exactly $n = 2$ `HeadDirectionCells`; one for leftward motion and one for rightward motion.

$$F_1(t) = \max(0, \text{sgn}(v_{1D}(t)))$$

$$F_2(t) = \max(0, \text{sgn}(-v_{1D}(t))) \tag{39}$$

### VelocityCells

`VelocityCells` are a subclass of `HeadDirectionCells` which encode the full velocity vector rather than the (normalised) head direction. In this sense they are similar to `HeadDirectionCells` but their firing rate will increase with the speed of the `Agent`.

In 2D their firing rate is given by:

$$F_i(t) = \frac{v_{2D}}{\sigma_v} \exp(\kappa_i \cos(\theta_v(t) - \theta_i)) \tag{40}$$

where $\theta_v(t)$ is the angle of the velocity vector $\mathbf{v}(t)$ anticlockwise from the x-direction and $\sigma_v$ is the likely speed scale of the `Agent` moving under random motion (this is chosen so the firing rate of the velocity cell before scaling is approximately O(1) Hz).

In 1D environments:

$$F_1(t) = \max\left(0, \frac{v_{1D}(t)}{\sigma_v + \mu_v}\right),$$
$$F_2(t) = \max\left(0, -\frac{v_{1D}(t)}{\sigma_v + \mu_v}\right) \tag{41}$$

where the addition of $\mu_v$ accounts for any bias in the motion.

## SpeedCell

A single cell encodes the scaled speed of the `Agent`

$$F(t) = \frac{\|\mathbf{v}(t)\|}{\sigma_v} \tag{42}$$

where, same as with the `VelocityCells`, $\sigma_v$ (or $\sigma_v + \mu_v$ in 1D) is the typical speed scale of the `Agent` moving under random motion giving these cells ad pre-scaled maximum firing rate of O(1) Hz.

## PhasePrecessingPlaceCells

`PhasePrecessingPlaceCells` (a subclass of `PlaceCells`) display a phenomena known as phase precession with respect to an underlying theta oscillation; within each theta cycle the firing rate of a place cell peaks at a phase dependent on how far through the place field the `Agent` has travelled. Specifically, as the `Agent` enters the receptive field the firing rate peaks at a late phase in the cycle and as the `Agent` leaves the receptive field the firing rate peaks at an early phase in the cycle, hence the name phase *precession*. Phase precession is implemented by modulating the spatial firing rate of `PlaceCells` with a phase precession factor, $F_i^\theta(t)$,

$$F_i(t) \leftarrow F_i(t) \cdot F_i^\theta(t), \tag{43}$$

which rises and falls each theta cycle, according to:

$$F_i^\theta(t) = 2\pi f_{VM}\left(\phi_\theta(t) \middle| \phi_i^*\left(\mathbf{x}(t), \dot{\mathbf{x}}(t)\right), \kappa_\theta\right). \tag{44}$$

This is a von Mises factor where $\phi_\theta(t) = 2\pi\nu_\theta t \mod 2\pi$ is the current phase of the $\nu_\theta$ Hz theta-rhythm and $\phi_i^*\left(\mathbf{x}(t), \hat{\mathbf{x}}(t)\right)$ is the current 'preferred' theta phase of a cell which is a function of it's position $\mathbf{x}(t)$ and direction of motion $\hat{\mathbf{x}}(t)$. This preferred phase is calculated by first establishing how far through a cells spatial receptive field the `Agent` has travelled along its current direction of motion;

$$d_i(\mathbf{x}(t), \hat{\mathbf{x}}(t)) = (\mathbf{x}(t) - \mathbf{x}_i) \cdot \hat{\mathbf{x}}(t), \tag{45}$$

and then mapping this to a uniform fraction $\beta_\theta$ of the range $[0, 2\pi]$;

$$\phi_i^*(t) = \pi - \beta_\theta \pi \frac{d_i(t)}{\sigma_i}. \tag{46}$$

$\sigma_i$ is the width of the cell at its boundary, typically defined as $\sigma_i = w_i$, except for `gaussian` place cells where the boundary is arbitrarily drawn at two standard deviations $\sigma_i = 2w_i$.

The intuition for this formula can be found by considering an `Agent` travelling straight through the midline of a circular 2D place field. As the `Agent` enters into the receptive field (at which point $(\mathbf{x}(t) - \mathbf{x}_i) \cdot \hat{\mathbf{x}}(t) = -\sigma_i$) the firing rate will peak at a theta phase of $\pi + \beta\pi$. This then precesses backwards as it passes through the field until the moment it leaves $((\mathbf{x}(t) - \mathbf{x}_i) \cdot \hat{\mathbf{x}}(t) = \sigma_i)$ when the firing rate peaks at a phase of $\pi - \beta\pi$. This generalises to arbitrary curved paths through 2D receptive fields. This

model has been used and validated before by *Jeewajee et al., 2014* . $\kappa_\theta$ determines the spread of the von Mises, i.e. how far from the preferred phase the cell is likely to fire.

## RandomSpatialNeurons

RandomSpatialNeurons provide spatially 'tuned' inputs for use in instances where PlaceCells, GridCells, BoundaryVectorCells etc. These neurons have smooth but, over long distances, random receptive fields (approximately) generated by sampling from a Gaussian process with a radial basis function kernel of lengthscale $l$ (default $l = 0.1$ m). The kernel is given by:

$$k(\mathbf{x}, \mathbf{x}') = \exp^{-\frac{d(\mathbf{x},\mathbf{x}')^2}{2l^2}} \tag{47}$$

where $d(\mathbf{x}, \mathbf{x}')$ is the environmental-distance between two points in the environment. This distance measure (same as used for PlaceCells, and VectorCells etc.) accounts for walls in the environment and so the receptive fields of these neurons are smooth everywhere except across walls (see Section 'Distance measures').

Firing rates are calculated as follows: At initialisation an array of target locations, at least as dense as the lengthscale, is sampled across the environment $\{\mathbf{x}_j\}$. For each neuron, $i$, $j$ target *values*, $[\tilde{F}_i]_{:}$, is sampled from the multivariate Normal distribution

$$[\tilde{F}_i]_{:} \sim \mathcal{N}(\mathbf{0}, \mathbf{K}) \tag{48}$$

where $\mathbf{K}$ is the covariance matrix with elements $K_{lm} = k(\mathbf{x}_l, \mathbf{x}_m)$. This creates a sparse set of locations, $\{\mathbf{x}_j\}$, and targets, $\tilde{F}_{ij}$, across the Environment: locations close to each other are likely to have similar targets (and hence similar firing rates) whereas locations far apart will be uncorrelated.

At inference time the firing rate at an arbitrary position in the Environment, $\mathbf{x}(t)$ (which will not neccesarily be one of the pre-sampled targets) is estimated by taking the mean of the targets weighted by the kernel function between the position and the target location:

$$F_i(\mathbf{x}(t)) = \frac{\sum_j k(\mathbf{x}(t), x_j)\tilde{F}_{i,j}}{\sum_j k(\mathbf{x}(t), x_j)} \tag{49}$$

This weighted average is a cheap and fast approximation to the true Bayesian Gaussian process which would require the inversion of the covariance matrix $\mathbf{K}$ at each time-step and which we find to be numerically unstable around exposed walls.

## FeedForwardLayer

FeedForwardLayer and NeuralNetworkNeurons are different from other RatInABox classes; their firing rates are not textitexplicitly determined by properties (position, velocity, head direction etc.) of their Agent but by the firing rates of a set of input layers (other ratinabox.Neurons). They allow users to create arbitrary and trainable 'function approximator' Neurons with receptive fields depending non-trivially on the states of one or many Agent(s).

Each FeedForwardLayer has a list of inputs $\{L_j\}_{j=1}^N$ which must be other ratinabox.Neurons subclasses (e.g. PlaceCells, BoundaryVectorCells, FeedForwardLayer). For input layer $j$ with $n_j$ neurons of firing rates $F_k^{L_j}(t)$ for $k \in [1, n_j]$, a weight matrix is initialised by drawing weights randomly $w_{ik}^{L_j} \sim \mathcal{N}(0, g/\sqrt{n_j})$ (for default weight intialisation scale $g = 1$). The firing rate of the $i^{th}$ Feed-ForwardLayer neuron is given by weighted summation of the inputs from all layers plus a bias term:

$$r_i(t) = \sum_{j=1}^N \sum_{k=1}^{n_j} w_{ik}^{L_j} F_k^{L_j}(t) + b_i \tag{50}$$

$$F_i(t) = \phi(r_i(t)) \tag{51}$$

where $\phi(x)$ is a potentially non-linear activation function defaulting to a linear identity function of unit gain. $b_i$ is a constant bias (default zero). A full list of available activations and their defining parameters

can be found in the utils.py file; these include ReLU, sigmoid, tanh, Retanh, softmax and linear (the default) functions or users can pass their own bespoke activation function.

Alongside $\phi(r_i(t))$ this layer also calculates and saves $\phi'(r_i(t))$ where $\phi'$ is the derivative of the activation function, a necessary quantity for many learning rules and training algorithms.

## NeuralNetworkNeurons

`NeuralNetworkNeurons` are a generalisation of `FeedForwardLayer`. Like `FeedForwardLayer` they are initialised with a list of inputs $\{L_j\}_{j=1}^{N}$. This class also recieves, at the point of initialisation, a neural network, `NN`. This can be any `pytorch.nn.module`. To calculate teh firing rate this class takes the firing rates of all input layers, concatenates them, and passes them through the neural network. The firing rate of the `NeuralNetworkNeurons` neuron is given by the activity of the neuron in the output layer of neural network:

$$F_i(t) = \mathrm{NN}_i(\underbrace{\vec{F}^{L_1}(t), \vec{F}^{L_2}(t), ...;}_{\text{inputs}} \underbrace{w}_{\text{weights}})$$

(52)

If no neural network is provided by the user a default network with two hidden ReLU layers of size 20 is used.

In order to be compatible with the rest of the RatInABox API the firing rate returned by this class is a numpy array, however, on each update the output of the `pytorch` neural network is additionally saved as a `torch` tensor. By accessing this tensor, users can take gradients back through the embedded neural network and train is as we demonstrate in *Figure 3e*.

In *Figure 3e* and an associated demo script a `NeuralNetworkNeurons` layer is initialised with $N = 1$ neuron/output. The inputs to the network come from a layer of 200 `GridCells`, ranging in grid scale from 0.2 m to 0.5 m. These are passed through a neural network with three hidden ReLU layers of size 100 and a linear readout. As the `Agent` randomly explores its `Environment` the network is trained with gradient descent to reduce the L2 error between the firing rate of the network and that of a 'target' rate map (a vector image of the letters 'RIAB'). We use gradient descent with momentum and a learning rate of $\eta = 0.002 \cdot \mathrm{dt}^2$ (which makes the total rate of learning time-step independent). Momentum is set to $\mu = (1 - \frac{dt}{\tau_{\text{et}}})$ where $\tau_{\text{et}}$ is the eligibility trace timescale of 10 s which smoothes the gradient descent, improving convergence. We find learning converges after approximately 2 hr and a good approximation of the target function is achieved.

## Tutorials and demonstrations

We provide numerous resources, some of which are listed here, to streamline the process of learning RatInABox. Next to each we describe the key features – which you may be interested in learning – covered by the resource.

- Github readme: Installing and importing RatInABox. Descriptions and diagrams of key features.
- Simple script: A minimal example of using RatInABox to generate and display data. Code duplicated below for convenience.
- Extensive script: A more detailed tutorial showing advanced data generation, and advanced plotting.
- Decoding position example: Data collection. Firing rate to position decoding. Data plotting.
- Conjunctive grid cells example: `GridCells` and `HeadDirectionCells` are combined with the function approximator `FeedForwardLayer` class to make head direction-selective grid cells (aka. conjunctive grid cells)
- Splitter cells example: Bespoke `Environment`, `Agent` and `Neurons` subclasses are written to make simple model of splitter cells.
- Successor features example: Loop-shaped `Environment` is constructed. Implementation of TD learning.
- Reinforcement Learning Example: A bespoke `ValueNeuron` subclass is defined. Implementation of TD learning. External 'non-random' control of `Agent` velocity.

- **Deep learning example**: Deep `NeuralNetworkNeurons` trained to approximate a target function. Bespoke `Neurons` subclass encoding a.png is written.
- **Actor-critic example**: Deep `NeuralNetworkNeurons` are used to implement the actor-critic algorithm in egocentric and allocentric action/representation spaces.
- **Path Integration Example**: Extensive use of `FeedForwardLayer` to build a deep multilayer network. Implementation of a local Hebbian learning rule.
- **List of plotting functions**: Lists and describes all available plotting functions.

In addition, scripts reproducing all figures in the GitHub readme and this paper are provided too. The code comments are nearly comprehensive and can be referenced for additional understanding where needed.

## A simple script

See the GitHub repository for instructions on how to install RatInABox. The following is a Python script demonstrating a very basic use-case.

Import RatInABox and necessary classes. Initialise a 2D `Environment`. Initialise an `Agent` in the `Environment`. Initialise some `PlaceCells`. Simulate for 20 s. Print table of times, position and firing rates. Plot the motion trajectory, the firing rate timeseries' and place cell rate maps.

```
# Import RatInABox
import ratinabox from ratinabox.Environment import Environment
from ratinabox.Agent import Agent
from ratinabox.Neurons import PlaceCells
import pandas as pd

# Run a very simple simulation
Env = Environment()
Ag = Agent(Env)
PCs = PlaceCells(Ag)
for i in range(int(20/Ag.dt)):
Ag.update()
PCs.update()

# Export data into a dataframe
pd.DataFrame(Ag.history)

# Plot data
Ag.plot_trajectory()
PCs.plot_rate_timeseries()
PCs.plot_rate_map()
```

## Table of default parameters

*Table 1* lists the RatInABox parameters and their default values. The 'Key' column give the key in a parameters dictionary which can be passed to each class upon initialisation. Any variables not present in the parameters dictionary at initialisation will be taken as default. For example, initialising an `Environment` of size 2 m (which is *not* the default size) and adding an `Agent` with a mean speed of 0.3ms⁻¹ (which is *not* the default size) would be done as follows:

```
import ratinaboxfrom ratinabox.Environment import Environment
from ratinabox.Agent import Agent

Env=Environment(params = "scale":2.0) # initialise non-default Environment
Ag=Agent(Env, params = "speed_mean":0.3) # initialise non-default Agent
```

**Table 1.** Default values, keys and allowed ranges for RatInABox parameters.

\* This parameter is passed as a kwarg to `Agent.update()` function, not in the input dictionary. \*\* This parameter is passed as a kwarg to `FeedForwardLayer.add_input()` when an input layer is being attached, not in the input dictionary.

| Parameter | Key | Description (unit) | Default | Acceptable range |
|---|---|---|---|---|
| | | `Environment()` | | |
| $D$ | `dimensionality` | Dimensionality of `Environment`. | `"2D"` | `["1D","2D"]` |
| Boundary conditions | `boundary_conditions` | Determines behaviour of `Agent` and `PlaceCells` at the room boundaries. | `"solid"` | `["solid", "periodic"]` |
| Scale, $s$ | `scale` | Size of the environment (m). | 1.0 | $\mathbb{R}^+$ |
| Aspect ratio, $a$ | `aspect` | Aspect ratio for rectangular 2D `Environments`; width = $sa$, height = $s$. | 1.0 | $\mathbb{R}^+$ |
| $dx$ | `dx` | Discretisation length used for plotting rate maps (m). | 0.01 | $\mathbb{R}^+$ |
| Walls | `walls` | A list of internal walls (not the perimeter walls) which will be added inside the `Environment`. More typically, walls will instead be added with the `Env.add_wall()` API (m). | [] | $N_{\text{walls}} \times 2 \times 2$-array/list |
| Boundary | `boundary` | Initialise non-rectangular `Environments` by passing in this list of coordinates bounding the outer perimeter (m). | None | $N_{\text{corners}} \times 2$-array/list |
| Holes | `holes` | Add multiple holes into the `Environment` by passing in a list of lists, each internal list contains coordinates (min 3) bounding the hole (m). | None | $N_{\text{holes}} \times \geq 3 \times 2$-array/list |
| Objects | `walls` | A list of objects inside the `Environment`. More typically, objects will instead be added with the `Env.add_object()` API (m). | [] | $N_{\text{objects}} \times 2$-array/list |
| | | `Agent()` | | |
| dt | `dt` | Time discretisation step size (s). | 0.01 | $\mathbb{R}^+$ |
| $\tau_v$ | `speed_coherence_time` | Timescale over which speed (1D or 2D) decoheres under random motion (s). | 0.7 | $\mathbb{R}^+$ |
| $\sigma_v$ (2D) $\mu_v$ (1D) | `speed_mean` | 2D: Scale Rayleigh distribution scale parameter for random motion in 2D. 1D: Normal distribution mean for random motion in 1D (ms⁻¹). | 0.08 | 2D: $\mathbb{R}^+$ 1D: $\mathbb{R}$ |
| $\sigma_v$ | `speed_std` | Normal distribution standard deviation for random motion in 1D (ms⁻¹). | 0.08 | $\mathbb{R}^+$ |
| $\tau_\omega$ | `rotational_velocity_ coherence_time` | Rotational velocity decoherence timescale under random motion (s). | 0.08 | $\mathbb{R}^+$ |

*Table 1 continued on next page*

*Table 1 continued*

| Parameter | Key | Description (unit) | Default | Acceptable range |
|---|---|---|---|---|
| $\sigma_\omega$ | `rotational_velocity_std` | Rotational velocity Normal distribution standard deviation (rad s⁻¹). | $2\pi/3$ | $\mathbb{R}^+$ |
| $\lambda_{\text{thig}}$ | `thigmotaxis` | Thigmotaxis parameter. | 0.5 | $0 < \lambda_{\text{thig}} < 1$ |
| $d_{\text{wall}}$ | `wall_repel_distance` | Wall range of influence (m). | 0.1 | $\mathbb{R}^+$ |
| s | `walls_repel_strength` | How strength walls repel the `Agent`. 0=no wall repulsion. | 1.0 | $\mathbb{R}_0^+$ |
| $k$ | `drift_to_random_strength_ratio*` | How much motion is dominated by the drift velocity (if present) relative to random motion. | 1.0 | $\mathbb{R}_0^+$ |
| | | `Neurons()` | | |
| $n$ | n | Number of neurons. | 10 | $\mathbb{Z}^+$ |
| $f_{\max}$ | `max_fr` | Maximum firing rate, see code for applicable cell types (Hz). | 1.0 | $\mathbb{R}$ |
| $f_{\min}$ | `min_fr` | Minimum firing rate, see code for applicable cell types (Hz). | 0.0 | $f_{\min} < f_{\max}$ |
| $\sigma_\eta$ | `noise_std` | Standard deviation of OU noise added to firing rates (Hz). | 0.0 | $\mathbb{R}^+$ |
| $\tau_\eta$ | `noise_coherence_time` | Timescale of OU noise added to firing rates (s). | 0.5 | $\mathbb{R}^+$ |
| Name | `name` | A name which can be used to identify a `Neurons` class. | `"Neurons"` | Any string |
| | | `PlaceCells()` | | |
| Type | `description` | Place cell firing function. | `"gaussian"` | `["gaussian", "gaussian_threshold", "diff_of_gaussians", "top_hat", "one_hot"]` |
| $w_i$ | `widths` | Place cell width parameter; can be specified by a single number (all cells have same width), or an array (each cell has different width) (m). | 0.2 | $\mathbb{R}^+$ |
| $\{\mathbf{x}_i^{\text{PC}}\}$ | `place_cell_centres` | Place cell locations. If `None`, place cells are randomly scattered (m). | `None` | None or array of positions (length $n$) |
| Wall geometry | `wall_geometry` | How place cells interact with walls. | `"geodesic"` | `["geodesic", "line_of_sight", "euclidean"]` |
| | | `GridCells()` | | |
| $\lambda_i$ | `gridscale` | Grid scales (m), or parameters for grid scale sampling distribution. | (0.5,1) | array-like or tuple |
| $\lambda_i$-dist | `gridscale_distribution` | The distribution from which grid scales are sampled, if they aren't manually provided as an array/list. | `"uniform"` | see `utils.distribution_sampler()` for list |
| $\theta_i$ | `orientation` | Orientations (rad), or parameters for orientation sampling distribution. | (0,2π) | array-like or tuple |

*Table 1 continued on next page*

*Table 1 continued*

| Parameter | Key | Description (unit) | Default | Acceptable range |
|---|---|---|---|---|
| $\theta_i$-dist | `orientation_distribution` | The distribution from which orientations are sampled, if they aren't manually provided as an array/list. | `"uniform"` | see `utils.distribution_sampler()` for list |
| $\phi_i$ | `phase_offset` | Phase offsets (rad), or parameters for phase offset sampling distribution. | $(0,2\pi)$ | array-like or tuple |
| $\phi_i$-dist | `phase_offset_distribution` | The distribution from which phase offsets are sampled, if they aren't manually provided as an array/list. | `"uniform"` | see `utils.distribution_sampler()` for list |
| Type | `description` | Grid cell firing function. | `"three_rectified_cosines"` | `["three_rectified_cosines","three_shifted_cosines"]` |
| | | `VectorCells()` | | |
| Reference frame | `reference_frame` | Whether receptive fields are defined in allo- or egocentric coordinate frames | `"allocentric"` | `["allocentric", "egocentric"]` |
| Arrangement protocol | `cell_arrangement` | How receptive fields are arranged in the environment. | `"random"` | `["random", "uniform_manifold", "diverging_manifold", function()]` |
| $d_i$ | `tuning_distance` | Tuning distances (m), or parameters for tuning distance sampling distribution. | $(0.0,0.3)$ | array-like or tuple |
| $d_i$-dist | `tuning_distance_distribution` | The distribution from which tuning distances are sampled, if they aren't manually provided as an array/list. | `"uniform"` | see `utils.distribution_sampler()` for list |
| $\sigma_{d,i}$ | `sigma_distance` | Distance tuning widths (m), or parameters for distance tuning widths distribution. (By default these give $\xi$ and $\beta$) | $(0.08,12)$ | array-like or tuple |
| $\sigma_{d,i}$-dist | `sigma_distance_distribution` | The distribution from which distance tuning widths are sampled, if they aren't manually provided as an array/list. "diverging" is an exception where distance tuning widths are an increasing linear function of tuning distance. | `"diverging"` | see `utils.distribution_sampler()` for list |
| $\phi_i$ | `tuning_angle` | Tuning angles (o), or parameters for tuning angle sampling distribution (degrees). | $(0.0,360.0)$ | array-like or tuple |
| $\phi_i$-dist | `tuning_angle_distribution` | The distribution from which tuning angles are sampled, if they aren't manually provided as an array/list. | `"uniform"` | see `utils.distribution_sampler()` for list |
| $\sigma_{\phi,i}$ | `sigma_angle` | Angular tuning widths (o), or parameters for angular tuning widths distribution (degrees). | $(10,30)$ | array-like or tuple |
| $\sigma_{\phi,i}$-dist | `sigma_angle_distribution` | The distribution from which angular tuning widths are sampled, if they aren't manually provided as an array/list. | `"uniform"` | see `utils.distribution_sampler()` for list |

*Table 1 continued on next page*

*Table 1 continued*

| Parameter | Key | Description (unit) | Default | Acceptable range |
|---|---|---|---|---|
| | | `BoundaryVectorCells()` | | |
| $d\theta$ | `dtheta` | Size of angular integration step (°). | 2.0 | $0 < d\theta << 360$ |
| | | `ObjectVectorCells()` | | |
| $t_i$ | `object_tuning_type` | Tuning type for object vectors, if `"random"` each OVC has preference for a random object type present in the environment | `"random"` | `"random"` or any-int or arrray-like |
| wall-behaviour | `walls_occlude` | Whether walls occlude objects behind them. | True | `bool` |
| `AgentVectorCells()` | | | | |
| Other agent, $j$ | `Other_Agent` | The `ratinabox.Agent` which these cells are selective for. | `None` | `ratinabox.Agent` |
| wall-behaviour | `walls_occlude` | Whether walls occlude `Agents` behind them. | True | `bool` |
| | | `FieldOfView[X]s()` for [X] ∈ [BVC,OVC,AVC] | | |
| $r_{\text{fov}}$ | `distance_range` | Radial extent of the field-of-view (m). | [0.02,0.4] | List of two distances |
| $\theta_{\text{fov}}$ | `angle_range` | Angular range of the field-of-view (°). | [0,75] | List of two angles |
| $\delta_{\text{fov}}^0$ | `spatial_resolution` | Resolution of the inner-most row of vector cells (m) | 0.02 | |
| $\beta$ | `beta` | Inverse gradient for how quickly receptie fields increase with distance (for `"diverging_manifold"` only) | 5 | $\mathbb{R}^+$ |
| Arrangement protocol | `cell_arrangement` | How the field-of-view receptive fields are constructed | `"diverging_manifold"` | `["diverging_manifold", "uniform_manifold"]` |
| | | `FeedForwardLayer()` | | |
| $\{L_j\}_{j=1}^N$ | `input_layers` | A list of `Neurons` classes which are upstream inputs to this layer. | [] | $N$-list of `Neurons` for $N \geq 1$ |
| Activation function | `activation_function` | Either a dictionary containing parameters of premade activation functions in `utils.activate()` or a user-define python function for bespoke activation function. | `{"activation": "linear"}` | See `utils.activate()` for full list |
| $g$ | `w_init_scale**` | Scale of random weight initialisation. | 1.0 | $\mathbb{R}^+$ |
| $b_i$ | `biases` | Biases, one per neuron (optional). | [0,....,0] | $\mathbb{R}^n$ |
| | | `NeuralNetworkNeurons()` | | |
| $\{L_j\}_{j=1}^N$ | `input_layers` | A list of `Neurons` classes which are upstream inputs to this layer. | [] | A list of `Neurons` |

*Table 1 continued*

| Parameter | Key | Description (unit) | Default | Acceptable range |
|---|---|---|---|---|
| NN | `NeuralNetworkModule` | The internal neural network function which maps inputs to outputs. If None a default ReLU networ kwith two-hidden layers of size 20 will be used. | None | Any `torch.nn.module` |
| | | `RandomSpatialNeurons()` | | |
| $l$ | `lengthscale` | Lengthscale of the Gaussian process kernel (m). | 0.1 | $\mathbb{R}^+$ |
| Wall geometry | `wall_geometry` | How distances are calculated and therefore how these cells interact with walls. | `"geodesic"` | `["geodesic"`, `"line_of_sight"`, `"euclidean"]` |
| | | `PhasePrecessingPlaceCells()` | | |
| $\nu_\theta$ | `theta_freq` | The theta frequency (Hz). | 10.0 | $\mathbb{R}^+$ |
| $\kappa_\theta$ | `kappa` | The phase precession breadth parameter. | 1.0 | $\mathbb{R}^+$ |
| $\beta_\theta$ | `beta` | The phase precession fraction. | 0.5 | $0.0 < \beta < 1.0$ |

## License

RatInABox is currently distributed under an MIT License, meaning users are permitted to use, copy, modify, merge publish, distribute, sublicense and sell copies of the software.

## Acknowledgements

We thank Tom Burns, Gastón Sivori, Peter Vincent and Colleen Gillon for helpful discussions and comments on the manuscript.

## Additional information

### Funding

| Funder | Grant reference number | Author |
|---|---|---|
| Wellcome | SRF 212281_Z_18_Z | William de Cothi |

The funders had no role in study design, data collection and interpretation, or the decision to submit the work for publication. For the purpose of Open Access, the authors have applied a CC BY public copyright license to any Author Accepted Manuscript version arising from this submission.

### Author contributions

Tom M George, Conceptualization, Software, Formal analysis, Validation, Investigation, Visualization, Methodology, Writing – original draft, Writing – review and editing; Mehul Rastogi, Software; William de Cothi, Claudia Clopath, Kimberly Stachenfeld, Caswell Barry, Supervision, Writing – review and editing

### Author ORCIDs

Tom M George (iD) http://orcid.org/0000-0002-4527-8810
Claudia Clopath (iD) https://orcid.org/0000-0003-4507-8648

### Decision letter and Author response

Decision letter https://doi.org/10.7554/eLife.85274.sa1
Author response https://doi.org/10.7554/eLife.85274.sa2

# Additional files

## Supplementary files
• MDAR checklist

## Data availability
Code is provided on the GitHub repository https://github.com/TomGeorge1234/RatInABox (*George, 2022*).

The following previously published dataset was used:

| Author(s) | Year | Dataset title | Dataset URL | Database and Identifier |
|---|---|---|---|---|
| Sargolini F | 2006 | Grid cell data Sargolini et al 2006 | https://doi.org/10.11582/2017.00019 | NIRD Research Data Archive, 10.11582/2017.00019 |

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

## Appendix 1

### 1.1 Figure details and parameter settings

A Jupyter script replicating *Figures 1–3* can be found at https://github.com/RatInABox-Lab/
RatInABox/blob/main/demos/paper_figures.ipynb. Unless stated below, parameters in all figures
take their default values listed in *Table 1*.

*Figure 1*: Panel (b): Place cells are of type `gaussian_threshold` with widths $w_i = 0.4$ m. Panel
(e) $\mu_v = 0.1$ and $\sigma_v = 0.2$.

*Figure 2*: Panel (a): Curve fitting is done using `scipy.optimize.curve_fit`. Panel (d): $dt = 100$
ms. Panel (e) `Agent.wall_repel_strength` = 2. Panel (e) uses *all* available datasets from *Sargolini
et al., 2006* to create the historgrams, as opposed to panel (a) which only uses one of the recordings.

*Figure 3*: Panel (a): 25 seconds of trajectory data from *Sargolini et al., 2006* is imported,
converted into metres, mean centred and then downsampled by 30 x (from 50 Hz to 1.66 Hz) before
being imported into a RatInABox `Agent`. Panel (c): All populations of vector cells had "`distance_
range`" = [0.05, 0.30], "`angle_range`" = [0,75] and "`spatial_resolution`"=0.015. Panel (e):
RatInABox for reinforcement learning experiment is described below. Panel (f): The average and
standard deviation over 1000 repeats is plotted. For the motion model this is taken for motion
updates of a default `Agent` in a default `Environment` (i.e. 2D with solid boundary conditions and
no additional walls). For the `numpy` matrix calculations the time taken does not include the time
taken to initialise the matrices.

### 1.2 Supplementary figures

In this section we demonstrate how RatInABox could be used in two simple experiments: neural
decoding and reinforcement learning. The intention is not to present novel scientific results but
rather to demonstrate the capability of RatInABox to facilitate novel scientific research in a variety of
fields. Additional demos beyond these two are given in the online repository and, as with all figures
in this paper, executable Jupyter scripts are provided to replicate all figures shown.

#### 1.2.1 *Appendix 1—figure 1* RatInABox for neural decoding

Jupyter script: https://github.com/RatInABox-Lab/RatInABox/blob/main/demos/paper_figures.
ipynb.

In this demonstration we study, using RatInABox, which type of spatially modulated cell type is
best for decoding position.

First we generate training and testing datasets. A set of `Neurons` ($n = N_{cells} = 20$) is initialised in
a 1 m square `Environment` containing a small barrier (*Appendix 1—figure 1*, top). A six minute
trajectory is simulated using the RatInABox random motion model to produce a dataset of inputs
$\{\mathbf{x}_i\}_{i=1}^{N_T}$ and targets $\{\mathbf{y}_i\}_{i=1}^{N_T}$:

$$\mathbf{x}_i = \vec{F}(\mathbf{x}(t_i)) \sim \mathcal{X} \subseteq \mathbb{R}^{N_{cells}} \tag{53}$$

$$\mathbf{y}_i = \mathbf{x}(t_i) \sim \mathcal{Y} \subseteq \mathbb{R}^2 \tag{54}$$

where $\mathbf{x}(t_i)$ is the position of the `Agent` at time $t_i$ and $\vec{F}$ is the firing rate of the neuronal population.
These data are split into training ($0 < t_i < 5$ mins, *Appendix 1—figure 1a* purple) and testing
($5 < t_i < 6$ mins, *Appendix 1—figure 1a* black) fractions. The goal of the decoder is to learn a
mapping $G : \mathcal{X} \to \mathcal{Y}$, from firing rates to positions.

To do this we use Gaussian Process Regression (GPR). GPR is a form of non-parameteric
regression where a prior is placed over the infinite-dimensional function space $P(G(\mathbf{x}))$ in the form of
its covariance kernel $C(\mathbf{x}, \mathbf{x}')$ and mean $\mu(\mathbf{x})$ (typically zero). This defines a prior on the targets in the
training set $\mathbf{Y} = (\mathbf{y}_1, \mathbf{y}_2, \mathbf{y}_3, ...)^\top$,

$$P(\mathbf{Y}) = \mathcal{N}(\mathbf{Y}; \mathbf{0}, \mathsf{C}), \tag{55}$$

where $\mathsf{C}_{ij} = C(\mathbf{x}_i, \mathbf{x}_j) + \sigma_\eta \delta_{ij}$ is a covariance matrix established over the data points. The second
term accounts for additive noise in the data function. This can be used to make an inference on
the posterior of the target for an unseen testing data point, $P(\mathbf{y}_{test}|\{\mathbf{x}_i\}^{train}, \{\mathbf{y}_i\}^{train}, \mathbf{x}_{test})$ – itself a

Gaussian – the mean of which is taken as the "prediction". A more comprehensive reference/tutorial on Gaussian Process Regression is given by *MacKay, 2003*.

We use a radial basis function (aka "squared exponential") kernel with width $l = l_0 \sqrt{N_{\text{cells}}}$ which scales with the expected size of the population vector ($\sim \sqrt{N_{\text{cells}}}$, we set $l_0 = \sqrt{20}$)

$$C(\mathbf{x}, \mathbf{x}') = \exp\left(-\frac{\|\mathbf{x} - \mathbf{x}'\|^2}{2l^2}\right) \tag{56}$$

and a small amount of target noise $\sigma_\eta = 1e - 10$. Note that the closest 'parameterised' analog to GPR with an RBF kernel is linear regression against Gaussian basis features of length scale $l$. Since the Gaussian is a non-linear function this means our regression prior is also non-linear function of firing rate (and therefore potential non-biologically plausible). We choose to optimise with the `sklearn.gaussian_process.GaussianProcessRegressor` package. Note we do not attempt to optimise the hyperparameters $l_0$ or $\sigma_\eta$ which one would probably do in a more rigorous experiment. RatInABox parameters are all default with the exception that the place cells are of type `gaussian_threshold` and width $w_i = 0.4$ m and the timestep is set to $dt = 50$ ms.

*Appendix 1—figure 1b* (lower) show the results over comparable sets of `PlaceCells`, `GridCells` and `BoundaryVectorCells`. Coloured dots show the prediction – mean of the posterior – of the GPR model "trained" on all points in the training dataset for that particular cell type. This is plotted on top of the true trajectory, shown in black. `PlaceCells` perform best achieving under 1 cm average decoding error, followed by `BoundaryVectorCells` then `GridCells` where the decoded position is visibly noisy.

Place cells outperform grid cells which outperform BVCs irrespective of how many cells are using in the basis feature set. More cells gives lower decoding error. Decoding errors in *Appendix 1—figure 1c* are smaller than would be expected if one decoded from equivalently sized populations of *real* hippocampal neurons. There are likely many reasons for this. Real neurons are noisy, communicate sparsely through spikes rather than rates and, most likely, jointly encode position and many other behaviourally relevant (or irrelevant) variables simultaneously. All of these factors could be straightforwardly incorporated into this analysis using existing RatInABox functionality.

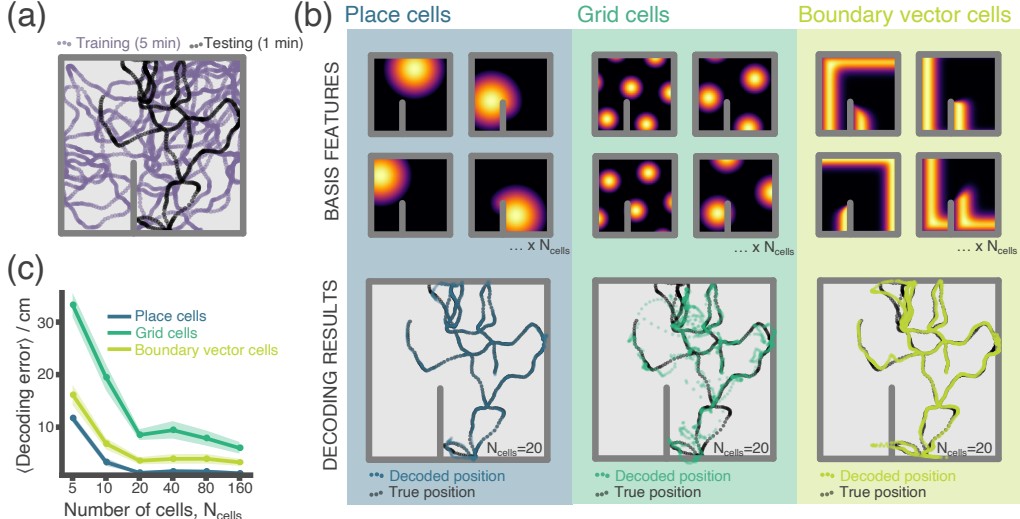

**Appendix 1—figure 1.** RatInABox used for a simple neural decoding experiment. (**a**) Training (5 min) and testing (1 min) trajectories are sampled in a 1 m square environment containing a small barrier. (**b**) The firing rates of a population of $N_{\text{cells}} = 20$ cells, taken over the training trajectory, are used to fit a Gaussian Process regressor model estimating position. This decoder is then used to decode position from firing rates on the the unseen testing dataset. Top row shows receptive field for 4 of the 20 cells, bottom row shows decoding estimate (coloured dots) against ground truth (black dots). The process is carried out independently for populations of place cells (left), grid cells (middle) and boundary vector cells (right). (**c**) Average decoding error against number of cells, note log scale. Error region shows the standard error in the mean over 15 random seeds. A jupyter script demonstrating this experiment is given in the codebase GitHub repository.

### 1.2.2 *Appendix 1—figure 2* RatInABox for reinforcement learning

Jupyter script: https://github.com/RatInABox-Lab/RatInABox/blob/main/demos/reinforcement_learning_example.ipynb.

In this example we demonstrate how RatInABox can be used in a reinforcement learning (RL) study. The goal is as follows: train an artificial `Agent` to explore a 2D `Environment` where a reward is hidden behind a wall. The `Agent` should become proficient at navigating around the wall and towards the reward from all locations within the `Environment`.

The core of our approach will rest on model-free RL where an `Agent` first learns a value function for a policy (a process known as "policy evaluation") and then uses this value function to define a new, improved policy ("policy improvement"). Iterating between these two procedures ("policy iteration") can result in convergence towards an optimal or near-optimal policy.

A core pillar of RatInABox is its continuous approach to modelling time and space. This continuity will require revising typical approaches to how the value function if defined, approximated and then learned, as well as how motion control (aka action selection, in discrete space) is performed. This is not a weakness, in fact we would argue it is one of the strengths. Once we are done, we are left with a formulation of model-free RL which bears much higher resemblance to biological navigation. Furthermore, since most of the complexities of feature encoding and motion control in continuous time and space are handled by RatInABox innately this "upgrade" comes almost for free.

### Policy evaluation

The value of a motion policy, $\pi$, is defined as the decaying sum (or integral in continuous time) of expected future rewards

$$\widehat{V}(t) \approx V^{\pi}(t) = \mathbb{E}\left[\frac{1}{\tau}\int_t^{\infty} e^{-\frac{t'-t}{\tau}} R(t')dt'\right] \tag{57}$$

where the expectation is taken over any stochasticity present in the current policy (i.e. how the `Agent` moves) and `Environment`/reward (although in this case both will be deterministic). This definition of value is temporally continuous. The key differences compared to the more common form – where value is written as a discrete sum of rewards over future timesteps – is that it is now a continuous integral over a reward density function and temporal discounting is done by exponentially decaying future reward over a time period $\tau$. The prefactor of $1/\tau$ is an optional constant of normalisation.

In order to learn the value function we define a new `ValueNeuron` class. The `ValueNeuron`, which is a subclass of `FeedForwardLayer`, recieves feedforward input from a set of features corresponding to `PlaceCells` scattered across the `Environment` with firing rates $\{\phi_i\}_{i=1}^{N_\phi=1000}$ where $\phi_i(\mathbf{x}) = F_i(\mathbf{x})$ is the firing rate of the $i^{\text{th}}$ place cell at location $\mathbf{x}$. This linear approximation to the value function can be written as

$$\widehat{V}(I(t); w) = \sum_{i=1}^{N} w_i \phi_i(t). \tag{58}$$

We can take the temporal derivative of *Equation (57)* and derive a consistency equation (analogous to the Bellman equation) satisfied by this value function. This naturally gives a temporal difference-style update rule which relies on "bootstrapping" (the current estimate of the value function is used in lieu of the true value function) to optimize the weights of the value function approximation. A good reference for continuous RL is *Doya, 2000* if readers wish to know more about deriving this learning rule.

$$\delta w_i(t) = \eta \left( R(t) + \tau \frac{d\widehat{V}(t)}{dt} - \widehat{V}(t) \right) e_i(t). \tag{59}$$

For now it suffices to observe that this learning rule is very similar to the temporally discrete TD-update rule. The first term in brackets represents the continuous analog of the temporal difference error (in fact, if you rediscretise using the Euler formula $\dot{V}(t) = \frac{V(t+dt)-V(t)}{dt}$ to replace the derivative, and set $dt = 1$, you will see they are identical). The second term is the 'eligibility trace' determining to which state – or basis feature – credit for the TD error should be assigned. Using an

eligibility trace is optional, and it could just be replaced with $\phi_i(t)$, however doing so aids stability of the learning. It is defined as:

$$e_i(t) = \frac{1}{\tau_e} \int_{-\infty}^{t} e^{-\frac{t-t'}{\tau_e}} \phi_i(t') dt'. \tag{60}$$

In total the newly defined ValueNeuron does three things, schematically laid out in *Appendix 1—figure 2*

1. It linearly summates its PlaceCell inputs, *Equation (58)*.
2. It stores and updates the eligibility traces, *Equation (60)*.
3. It implements the learning rule, *Equation (59)*, which requires access to the reward density function $R(t)$, the eligibility traces $e_i(t)$, its firing rate $\widehat{V}(t)$ and the temporal derivative of its firing rate $\frac{d\widehat{V}(t)}{dt}$.

We use a temporal discount horizon of $\tau = 10$ s and an eligibility trace timescale of $\tau_e = 5$ s. Input features are a set of $N_\phi = 1000$ `PlaceCells` of random widths uniformly sampled from 0.04 m to 0.4 m (*Appendix 1—figure 2b*). The reward density function is taken to be the firing rate of a single `PlaceCell` positioned behind the wall of type top_hat and width 0.2 m (*Appendix 1—figure 2c*). The learning rate is set to $\eta = 1e - 4$.

## Policy improvement

Now we have a neuron capable of learning the value function under its current policy ("policy evaluation"). We want to use this to improve the policy ("policy improvement") towards an optimal one. To do this we will exploit the "drift velocity" feature (see Section 'External velocity control'). We set the drift velocity to be 3 times the mean velocity in the direction of steepest ascent of the value function.

$$\mathbf{v}_{\text{drift}}(t) = 3\sigma_v \widehat{\nabla}_\mathbf{x} \widehat{V}(\mathbf{x}(t)). \tag{61}$$

This way the `Agent` is encouraged to move towards regions of higher and higher value. Note that calculating this gradient is a local calculation and can be done on-the-fly by the `Agent`, as it locomotes. This method of value ascent is essentially a continuous analog of a similar algorithm, "greedy policy optimization", used in discrete action spaces.

## Policy iteration

Learning is done in batches of 8 episodes each. An episode consists of the `Agent` being reset to a random location in the `Environment` and left to explore. The episode ends when the `Agent` gets close to the reward *or* times out (60 seconds). At the start of each batch the current value function is copied and cached - this cached version is used, but not updated, to determine the drift velocity in *Equation (61)* for the duration of the next batch. Varying the strength of the drift bias relative to the random motion allows us to control the trade of between exploration and exploitation. Scheduling goes as follows: initially the drift_to_random_strength_ratio is set to $k = 0.1$ (i.e. mostly random exploration). On each successful episode which did not end in a timeout, this is increased by 10% up to a maximum of $k = 1$ (approximately equal contributions of random and drift motions).

## Results

Initially the input weights to the `ValueNeuron` are drawn randomly $w_i \sim \mathcal{N}(0, \frac{1}{\sqrt{N_\phi}})$ and therefore the value map (and `Agent` motion) is random (*Appendix 1—figure 2d*, left). After 10 batches (80 episodes) the `Agent` has successfully learnt a near-optimal value function showing high-value in and near to the corridor, and low values elsewhere. This allows it to rapidly navigate towards the reward, avoiding the obstructing wall, from all locations in the `Environment` (*Appendix 1—figure 2d*, middle, colourmap) (*Appendix 1—figure 2d*, middle and *Figure 3b*).

By virtue of using continuous action control in continuous space, the trajectories of the trained `Agent` look highly realistic compared to typical gridworld RL. Since `PlaceCells` in RatInABox interact adaptively with the `Environment`, when a small gap is created at the base of the obstructing wall the receptive fields of `PlaceCells` near this gap "spill" through. This causes an instantaneous

update to the percieved value function and therefore policy allowing the `Agent` to immediately find a short cut to the reward with no additional training, a form of zero-shot learning (***Appendix 1—figure 2d***, right).

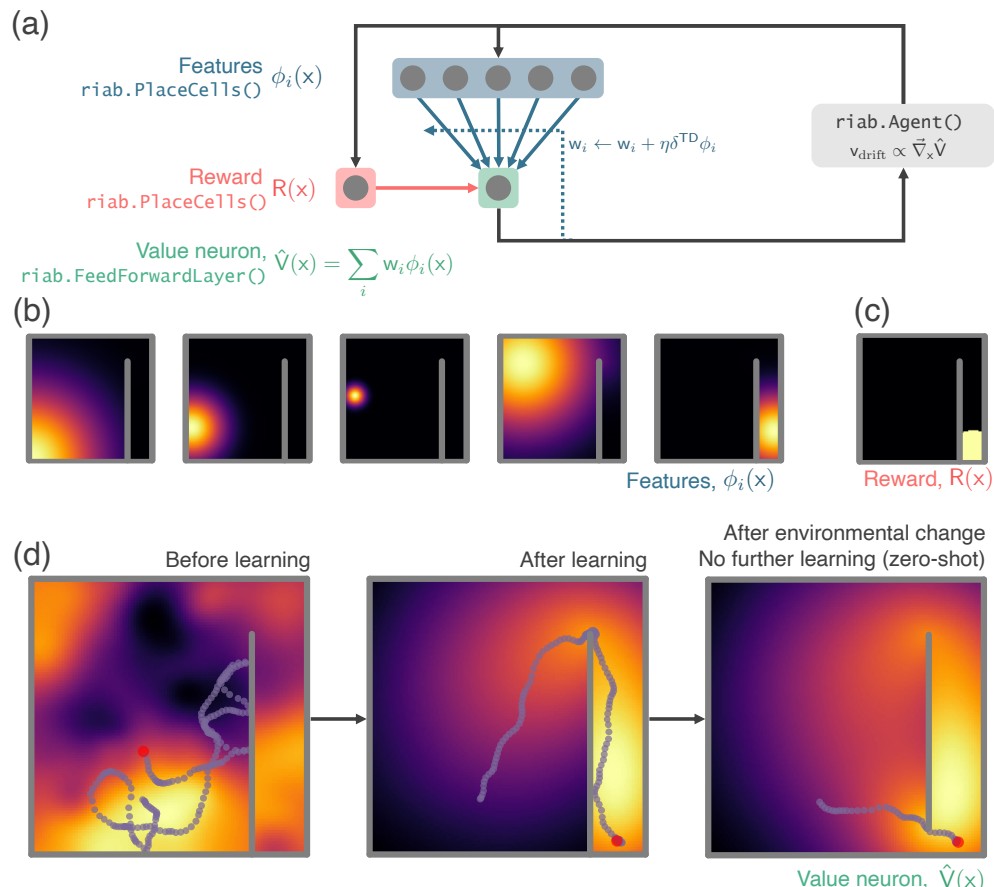

**Appendix 1—figure 2.** RatInABox used in a simple reinforcement learning project. (**a**) A schematic of the 1 layer linear network. Using a simple model-free policy iteration algorithm the `Agent`, initially moving under a random motion policy, learns to approach an optimal policy for finding a reward behind a wall. The policy iteration algorithm alternates between (left) calculating the value function using temporally continuous TD learning and (right) using this to define an improved policy by setting the drift velocity of the Agent to be proportional to the gradient of the value function (a roughly continuous analog for the $\epsilon$-greedy algorithm). (**b**) 1000 `PlaceCells` act as a continuous feature basis for learning the value function. (**c**) The reward is also a (top-hat) `PlaceCell`, hidden behind the obstructing wall. (**d**) A `ValueNeuron` (a bespoke `Neurons` subclass defined for this demonstration) estimates the policy value function as a linear combination of the basis features (heatmap) and improves this using TD learning. After learning the `Agent` is able to accurately navigate around the wall towards the reward (middle). Because `PlaceCells` in RatInABox are continuous and interact adaptively with the `Environment` when a small gap is opened in the wall place fields corresponding to place cells near this gap automatically bleed through it, and therefore so does the value function. This allows the `Agent` to find a shortcut to the reward with zero additional training. A jupyter script replicating this project is given in the demos folder GitHub repository.

