## [Editor Report]

RatInABox is a new python library for generating synthetic behavioral and neural data (many functional cell types) that is a highly important contribution to computational neuroscience. Critically, the authors have gone beyond the generally accepted practice with their well-written paper, documented and verified code. They show compelling evidence of its utility and usability, and this is sure to be an influential paper with implications for developing new theories and methods for joint neural and behavioral analysis beyond the navigation field.

---

## [Decision Letter]

**Decision letter after peer review:**

Thank you for submitting your article "RatInABox: A toolkit for modelling locomotion and neuronal activity in continuous environments" for consideration by *eLife*. Your article has been reviewed by 3 peer reviewers, one of whom is a member of our Board of Reviewing Editors, and the evaluation has been overseen by Michael Frank as the Senior Editor. The following individual involved in review of your submission has agreed to reveal their identity: Antonio Fernandez-Ruiz (Reviewer #2).

Essential revisions:

Overall the reviewers are positive about this work and have a few suggestions to improve the presentation and some questions. Here are the essential revisions we ask you to consider, and the detailed reviews are below for your reference.

1) We would like to see a better discussion of limitations in the Discussion, and some more staging of why this is useful in the Introduction. Specifically, please consider these comments:

R2: The manuscript also lacks a description of the limitations of the approach. The authors should clarify what types of experimental data that can and cannot be modelled here, the limitations of experimental estimates from the models, and more importantly, what testable predictions of yet unknown results can be generated by this model, beyond replicating what is already known. For example, can RatInABox be used to model episodic coding in the hippocampus (e.g., splitter cells), as task states can often be latent and non-Markovian. Can it be used to model the "remapping" of place cells (and other cell types) in response to changes in the environment?

R1: Presentation: Some discussion of case studies in Introduction might address the above point on impact. It would be useful to have more discussion of how general the software is, and why the current feature set was chosen. For example, how well does RatInABox deal with environments of arbitrary shape? T-mazes? It might help illustrate the tool's generality to move some of the examples in supplementary figure to main text – or just summarize them in a main text figure/panel.

2) It would be great to address limitations on how the environment and cells can be modelled currently. Namely, how the cells can be modified to study influences of other internal (latent) states, such as reward or abstract coding, and dynamics. And please clarify what types of experimental data that can and cannot be modeled, the limitations of experimental estimates from the models, and importantly, what testable predictions of yet unknown results can be generated by this model, beyond replicating what is already known. Specifically, please consider these comments from:

R3: Right now the code only supports X,Y movements, but Z is also critical and opens new questions in 3D coding of space (such as grid cells in bats, etc). Many animals effectively navigate in 2D, as a whole, but they certainly make a large number of 3D head movements, and modeling this will become increasingly important and the authors should consider how to support this.

– What about other environments that are not "Boxes" as in the name – can the environment only be a Box, what about a circular environment? Or Bat flight? This also has implications for the velocity of the agent, etc. What are the parameters for the motion model to simulate a bat, which likely has a higher velocity than a rat?

– What if a place cell is not encoding place but is influenced by reward or encodes a more abstract concept? Should a PlaceCell class inherit from an AbstractPlaceCell class, which could be used for encoding more conceptual spaces? How could their tool support this?

R2: The manuscript also lacks a description of the limitations of the approach. The authors should clarify what types of experimental data that can and cannot be modelled here, the limitations of experimental estimates from the models, and more importantly, what testable predictions of yet unknown results can be generated by this model, beyond replicating what is already known. For example, can RatInABox be used to model episodic coding in the hippocampus (e.g., splitter cells), as task states can often be latent and non-Markovian. Can it be used to model the "remapping" of place cells (and other cell types) in response to changes in the environment?

*Reviewer #2 (Recommendations for the authors):*

– With a few exceptions, the manuscripts lacks a comparison between the RatInABox and previous approaches to simulate behavioral and electrophysiological data. Such comparison can be added to the Discussion and will help many readers to appreciate the novelty and capabilities of this toolbox.

– An important strength of the toolbox is its capability to simulate with ease realistic animal exploratory patterns. In comparison, the simulation of electrophysiological cell responses is more simplistic. It would be useful to better describe the assumptions and limitations taken in the simulations of cell types and briefly compare them with the well-known experimental evidence. For example, place fields are pre-determined by the model's parameters and completely stable. However, it is well known that place fields developed with experience (e.g., change locations and sizes, pop in/out). The paper claims that it can "concurrently simulate neuronal activity data" in "configurable" environments. It should be clarified if this model can capture the developmental part of place cells, or only the "stable" state. As another example, the authors showed that the decoding error of Agent position is almost 0 with >= 20 place cells (Figure S1c), which is significantly less than that using real neural data. At least, there should be some discussion of where this difference may arise. Can noise be added to cell tuning curves? E.g. in the form of out-of-field spikes or trial-by-trial variability.

– The manuscript also lacks a description of the limitations of the approach. The authors should clarify what types of experimental data that can and cannot be modelled here, the limitations of experimental estimates from the models, and more importantly, what testable predictions of yet unknown results can be generated by this model, beyond replicating what is already known. For example, can RatInABox be used to model episodic coding in the hippocampus (e.g., splitter cells), as task states can often be latent and non-Markovian. Can it be used to model the "remapping" of place cells (and other cell types) in response to changes in the environment?

---

## [Author Response]

Essential revisions:Overall the reviewers are positive about this work and have a few suggestions to improve the presentation and some questions. Here are the essential revisions we ask you to consider, and the detailed reviews are below for your reference.1) We would like to see a better discussion of limitations in the Discussion, and some more staging of why this is useful in the Introduction. Specifically, please consider these comments:

We now openly discuss the limitations of the software package in the Discussion on page 9 in a new section of text as follows:

“Our package is not the first to model neural data [37,38,39] or spatial behaviour [40,41], yet it distinguishes itself by integrating these two aspects within a unified, lightweight framework. The modelling approach employed by RatInABox involves certain assumptions:

1. It does not engage in the detailed exploration of biophysical [37,39] or biochemical[38] aspects of neural modelling, nor does it delve into the mechanical intricacies of joint and muscle modelling [40,41]. While these elements are crucial in specific scenarios, they demand substantial computational resources and become less pertinent in studies focused on higher-level questions about behaviour and neural representations.

2. A focus of our package is modelling experimental paradigms commonly used to study spatially modulated neural activity and behaviour in rodents. Consequently, environments are currently restricted to being two-dimensional and planar, precluding the exploration of three-dimensional settings. However, in principle, these limitations can be relaxed in the future.

3. RatInABox avoids the oversimplifications commonly found in discrete modelling, predominant in reinforcement learning [22,23], which we believe impede its relevance to neuroscience.

4. Currently, inputs from different sensory modalities, such as vision or olfaction, are not explicitly considered. Instead, sensory input is represented implicitly through efficient allocentric or egocentric representations. If necessary, one could use the RatInABox API in conjunction with a third-party computer graphics engine to circumvent this limitation.

5. Finally, focus has been given to generating synthetic data from steady-state systems. Hence, by default, agents and neurons do not explicitly include learning, plasticity or adaptation. Nevertheless we have shown that a minimal set of features such as parameterised function-approximator neurons and policy control enable a variety of experience-driven changes in behaviour the cell responses [42, 43] to be modelled within the framework.”

And have also added a new opening paragraph to better stage why RatInABox is useful (and computational modelling toolkits in general) to *eLife*’s broad audience:

“Computational modelling provides a means to understand how neural circuits represent the world and influence behaviour, interfacing between experiment and theory to express and test how information is processed in the brain. Such models have been central to understanding a range of neural mechanisms, from action potentials [1] and synaptic transmission between neurons [2], to how neurons represent space and guide complex behaviour [3,4,5,6,7]. Relative to empirical approaches, models can offer considerable advantages, providing a means to generate large amounts of data quickly with limited physical resources, and are a precise means to test and communicate complex hypotheses. To fully realise these benefits, computational modelling must be accessible and standardised, something which has not always been the case.”

R2: The manuscript also lacks a description of the limitations of the approach. The authors should clarify what types of experimental data that can and cannot be modelled here, the limitations of experimental estimates from the models, and more importantly, what testable predictions of yet unknown results can be generated by this model, beyond replicating what is already known.

In the text described above and added to the discussion, we discuss the assumptions and limitations of RatInABox. Briefly these are: no low-level biophysical, biochemical, joint or muscle-level modelling, no 3D or higher dimensional / abstract environments, no discretised environments, no explicit sensory inputs (e.g. vision or olfaction).

Most of these limitations are, as we hope to have made clear in the manuscript, features not bugs. They were explicit choices to keep the RatInABox framework lightweight and easy to understand. We wanted to avoid “feature creep” and repeating work done by others. Interfacing RatInABox with other toolkits or exploiting the existing RatInABox API to add new features could overcome these limitations and we will happily consider these contributions as/when they are made.

Regarding “how the cells can be modified to study influences of other internal (latent) states, such as reward or abstract coding, and dynamics”*:* Briefly (and expanded on in the individual responses below), in a series of new tutorials we demonstrate how RatInABox can be used to model:

– Splitter cells^[1]^ (i.e. where a latent factor determines cell firing dynamics)

– Conjunctive cells (where multiple primitive cell classes combine to give mixed selective cells e.g. jointly selective to speed and position)

– Reinforcement learning^[2,3]^ (where reward is explicitly encoded and used to “train” representations and guide behaviour)

– Successor features^[4]^ (where cell representations are dependent on statistics of the agents trajectory).

To name a few. As such, the following text has been modified in the case studies, section 3.1.

“Additional tutorials, not described here but available online, demonstrate how RatInABox can be used to model splitter cells, conjunctive grid cells, biologically plausible path integration, successor features, deep actor-critic RL, whisker cells and more. Despite including these examples we stress that they are not exhaustive. RatInABox provides the framework and primitive classes/functions from which highly advanced simulations such as these can be built”

Furthermore, function-approximator neurons (such as the existing FeedForwardLayer, or the new NeuralNetworkNeurons) can be used to model dynamic systems either by (i) setting the input to a neuron layer as itself (a recurrent input) or (ii) setting the function approximator to be dynamic, e.g. a recurrent neural network. The following text has been added to section 2:

“Naturally, function-approximator Neurons can be used to model how neural populations in the brain communicate, how neural representations are learned or, in certain cases, neural dynamics. In an online demo we show how grid cells and head direction cells can be easily combined using a FeedForwardLayer to create head-direction selective grid cells (aka. conjunctive grid cells [10]). In Figure 3d and associated demo GridCells provide input to a NeuralNetworkNeuron which is then trained, on data generated during exploration, to have a highly complex and non-linear receptive field. Function-approximator Neurons can themselves be used as inputs to other function-approximator Neurons allowing multi-layer and/or recurrent networks to be constructed and studied.”

Finally, we already know that “testable predictions of yet unknown results can be generated by this model” as a few early citations of the package^[8,9,10]^ (which have been added to the manuscript) demonstrate. However, primarily we think of RatInABox as a generator of models rather than a model itself, therefore making testable predictions is not the primary contribution and might seem out of place. We have modified the following sentence in the discussion to clarify this:

“Its user-friendly API, inbuilt data-plotting functions and general yet modular feature set mean it is well placed empower a wide variety of users to more rapidly build, train and validate models of hippocampal function [25] and spatial navigation [26], accelerating progress in the field.”

As well as the final paragraph of the discussion which was added to say:

“In conclusion, while no single approach can be deemed the best, we believe that RatInABox’s unique positioning makes it highly suitable for normative modelling and NeuroAI. We anticipate that it will complement existing toolkits and represent a significant contribution to the computational neuroscience toolbox.”

To clarify that normative modelling and NeuroAI are two key areas we anticipate RatInABox making contributions.

R1: Presentation: Some discussion of case studies in Introduction might address the above point on impact. It would be useful to have more discussion of how general the software is, and why the current feature set was chosen. For example, how well does RatInABox deal with environments of arbitrary shape? T-mazes? It might help illustrate the tool's generality to move some of the examples in supplementary figure to main text – or just summarize them in a main text figure/panel.

Thank you for this question. Since the initial submission of this manuscript RatInABox has been upgraded and environments have become substantially more “general”. Environments can now be of arbitrary shape (including T-mazes), boundaries can be curved, they can contain holes and can also contain objects (0-dimensional points which act as visual cues). A few examples are showcased in the updated figure 1 panel e.

To further illustrate the tools generality beyond the structure of the environment we continue to summarise the reinforcement learning example (Figure 3e) and neural decoding example in section 3.1. In addition to this we have added three new panels into figure 3 highlighting new features which, we hope you will agree, make RatInABox significantly more powerful and general and satisfy your suggestion of clarifying utility and generality in the manuscript directly. These include:

– Figure 3b: Demonstration of this ability to control policy with a general control signal allowing users to generate arbitrarily complex motion trajectories

– Figure 3c: New cell classes use egocentric boundary-, object- and agent-vector cells to efficiently encode what is in the agents “field of view”.

– Figure 3d: Whilst most neurons have static receptive fields (e.g. place cells) we also provide two neuron classes whose receptive fields are defined by a parameterized function receiving the firing rate of other RatInABox neurons as inputs. In this example, grid cells are mapped through a feedforward neural network and trained to represent an arbitrary receptive field (the letter “riab”). These classes allow the construction of quite general neurons (a point we endeavour to stress in the manuscript) as well as the ability to study learning processes in the brain.

On the topic of generality, we wrote the manuscript in such a way as to demonstrate how the rich variety of ways RatInABox can be used without providing an exhaustive list of potential applications. For example, RatInABox *can* be used to study neural decoding and it *can* be used to study reinforcement learning but not because it was purpose built with these use-cases in mind. Rather because it contains a set of core tools designed to support spatial navigation and neural representations in general. For this reason we would rather keep the demonstrative examples as supplements and implement your suggestion of further raising attention to the large array of tutorials and demos provided on the GitHub repository by modifying the final paragraph of section 3.1 to read:

“Additional tutorials, not described here but available online, demonstrate how RatInABox can be used to model splitter cells, conjunctive grid cells, biologically plausible path integration, successor features, deep actor-critic RL, whisker cells and more. Despite including these examples we stress that they are not exhaustive. RatInABox provides the framework and primitive classes/functions from which highly advanced simulations such as these can be built.”

2) It would be great to address limitations on how the environment and cells can be modelled currently. Namely, how the cells can be modified to study influences of other internal (latent) states, such as reward or abstract coding, and dynamics. And please clarify what types of experimental data that can and cannot be modeled, the limitations of experimental estimates from the models, and importantly, what testable predictions of yet unknown results can be generated by this model, beyond replicating what is already known.

In the text described above and added to the discussion, we discuss the assumptions and limitations of RatInABox. Briefly these are: no low-level biophysical, biochemical, joint or muscle-level modelling, no 3D or higher dimensional / abstract environments, no discretised environments, no explicit sensory inputs (e.g. vision or olfaction).

Most of these limitations are, as we hope to have made clear in the manuscript, features not bugs. They were explicit choices to keep the RatInABox framework lightweight and easy to understand. We wanted to avoid “feature creep” and repeating work done by others. Interfacing RatInABox with other toolkits or exploiting the existing RatInABox API to add new features could overcome these limitations and we will happily consider these contributions as/when they are made.

Regarding “how the cells can be modified to study influences of other internal (latent) states, such as reward or abstract coding, and dynamics”*:* Briefly (and expanded on in the individual responses below), in a series of new tutorials we demonstrate how RatInABox can be used to model:

– Splitter cells^[1]^ (i.e. where a latent factor determines cell firing dynamics)

– Conjunctive cells (where multiple primitive cell classes combine to give mixed selective cells e.g. jointly selective to speed and position)

– Reinforcement learning^[2,3]^ (where reward is explicitly encoded and used to “train” representations and guide behaviour)

– Successor features^[4]^ (where cell representations are dependent on statistics of the agents trajectory).

To name a few. As such, the following text has been modified in the case studies, section 3.1:

“Additional tutorials, not described here but available online, demonstrate how RatInABox can be used to model splitter cells, conjunctive grid cells, biologically plausible path integration, successor features, deep actor-critic RL, whisker cells and more. Despite including these examples we stress that they are not exhaustive. RatInABox provides the framework and primitive classes/functions from which highly advanced simulations such as these can be built”

Furthermore, function-approximator neurons (such as the existing FeedForwardLayer, or the new NeuralNetworkNeurons) can be used to model dynamic systems either by (i) setting the input to a neuron layer as itself (a recurrent input) or (ii) setting the function approximator to be dynamic, e.g. a recurrent neural network. The following text has been added to section 2:

“Naturally, function-approximator Neurons can be used to model how neural populations in the brain communicate, how neural representations are learned or, in certain cases, neural dynamics. In an online demo we show how grid cells and head direction cells can be easily combined using a FeedForwardLayer to create head-direction selective grid cells (aka. conjunctive grid cells [10]). In Figure 3d and associated demo GridCells provide input to a NeuralNetworkNeuron which is then trained, on data generated during exploration, to have a highly complex and non-linear receptive field. Function-approximator Neurons can themselves be used as inputs to other function-approximator Neurons allowing multi-layer and/or recurrent networks to be constructed and studied.”

Finally, we already know that “testable predictions of yet unknown results can be generated by this model” as a few early citations of the package^[8,9,10]^ (which have been added to the manuscript) demonstrate. However, primarily we think of RatInABox as a generator of models rather than a model itself, therefore making testable predictions is not the primary contribution and might seem out of place. We have modified the following sentence in the discussion to clarify this:

“Its user-friendly API, inbuilt data-plotting functions and general yet modular feature set mean it is well placed empower a wide variety of users to more rapidly build, train and validate models of hippocampal function [25] and spatial navigation [26], accelerating progress in the field.”

As well as the final paragraph of the discussion which was added to say:

“In conclusion, while no single approach can be deemed the best, we believe that RatInABox’s unique positioning makes it highly suitable for normative modelling and NeuroAI. We anticipate that it will complement existing toolkits and represent a significant contribution to the computational neuroscience toolbox.”

To clarify that normative modelling and NeuroAI are two key areas we anticipate RatInABox making contributions.

Specifically, please consider these comments from:R3: Right now the code only supports X,Y movements, but Z is also critical and opens new questions in 3D coding of space (such as grid cells in bats, etc). Many animals effectively navigate in 2D, as a whole, but they certainly make a large number of 3D head movements, and modeling this will become increasingly important and the authors should consider how to support this.

Agents now have a dedicated head direction variable (before head direction was just assumed to be the normalised velocity vector). By default this just smoothes and normalises the velocity but, in theory, could be accessed and used to model more complex head direction dynamics. This is described in the updated methods section.

In general, we try to tread a careful line. For example we embrace certain aspects of physical and biological realism (e.g. modelling environments as continuous, or fitting motion to real behaviour) and avoid others (such as the biophysics/biochemisty of individual neurons, or the mechanical complexities of joint/muscle modelling). It is hard to decide where to draw but we have a few guiding principles:

1. RatInABox is most well suited for normative modelling and neuroAI-style probing questions at the level of behaviour and representations. We consciously avoid unnecessary complexities that do not directly contribute to these domains.

2. Compute: To best accelerate research we think the package should remain fast and lightweight. Certain features are ignored if computational cost outweighs their benefit.

3. Users: If, and as, users require complexities e.g. 3D head movements, we will consider adding them to the code base.

For now we believe proper 3D motion is out of scope for RatInABox. Calculating motion near walls is already surprisingly complex and to do this in 3D would be challenging. Furthermore all cell classes would need to be rewritten too. This would be a large undertaking probably requiring rewriting the package from scratch, or making a new package RatInABox3D (BatInABox?) altogether, something which we don’t intend to undertake right now. One option, if users really needed 3D trajectory data they could quite straightforwardly simulate a 2D Environment (X,Y) and a 1D Environment (Z) independently. With this method (X,Y) and (Z) motion would be entirely independent which is of unrealistic but, depending on the use case, may well be sufficient.

Alternatively, as you said that many agents effectively navigate in 2D but show complex 3D head and other body movements, RatInABox could interface with and feed data downstream to other softwares (for example Mujoco^[11]^) which specialise in joint/muscle modelling. This would be a very legitimate use-case for RatInABox.

We’ve flagged all of these assumptions and limitations in a new body of text added to the discussion:

“Our package is not the first to model neural data [37, 38, 39] or spatial behaviour [40, 41], yet it distinguishes itself by integrating these two aspects within a unified, lightweight framework. The modelling approach employed by RatInABox involves certain assumptions:

1. It does not engage in the detailed exploration of biophysical [37, 39] or biochemical [38] aspects of neural modelling, nor does it delve into the mechanical intricacies of joint and muscle modelling[40, 41]. While these elements are crucial in specific scenarios, they demand substantial computational resources and become less pertinent in studies focused on higher-level questions about behaviour and neural representations.

2. A focus of our package is modelling experimental paradigms commonly used to study spatially modulated neural activity and behaviour in rodents. Consequently, environments are currently restricted to being two-dimensional and planar, precluding the exploration of three-dimensional settings. However, in principle, these limitations can be relaxed in the future.

3. RatInABox avoids the oversimplifications commonly found in discrete modelling, predominant in reinforcement learning [22, 23], which we believe impede its relevance to neuroscience.

4. Currently, inputs from different sensory modalities, such as vision or olfaction, are not explicitly considered. Instead, sensory input is represented implicitly through efficient allocentric or egocentric representations. If necessary, one could use the RatInABox API in conjunction with a third-party computer graphics engine to circumvent this limitation.

5. Finally, focus has been given to generating synthetic data from steady-state systems. Hence, by default, agents and neurons do not explicitly include learning, plasticity or adaptation. Nevertheless we have shown that a minimal set of features such as parameterised function-approximator neurons and policy control enable a variety of experience-driven changes in behaviour the cell responses [42, 43] to be modelled within the framework.

In conclusion, while no single approach can be deemed the best, we believe that RatInABox’s unique positioning makes it highly suitable for normative modelling and NeuroAI. We anticipate that it will complement existing toolkits and represent a significant contribution to the computational neuroscience toolbox.”

– What about other environments that are not "Boxes" as in the name – can the environment only be a Box, what about a circular environment? Or Bat flight? This also has implications for the velocity of the agent, etc. What are the parameters for the motion model to simulate a bat, which likely has a higher velocity than a rat?

Thank you for this question. Since the initial submission of this manuscript RatInABox has been upgraded and environments have become substantially more “general”. Environments can now be of arbitrary shape (including circular), boundaries can be curved, they can contain holes and can also contain objects (0-dimensional points which act as visual cues). A few examples are showcased in the updated figure 1 panel e.

Whilst we don’t know the exact parameters for bat flight users could fairly straightforwardly figure these out themselves and set them using the motion parameters as shown in the table in Author response image 1. We would guess that bats have a higher average speed (speed_mean) and a longer decoherence time due to increased inertia (speed_coherence_time), so the following code might roughly simulate a bat flying around in a 10 x 10 m environment. The table (6.4 Table of default parameters) shows all Agent parameters which can be set to vary the random motion model.

**Author response image 1. sa2fig1:** 

– What if a place cell is not encoding place but is influenced by reward or encodes a more abstract concept? Should a PlaceCell class inherit from an AbstractPlaceCell class, which could be used for encoding more conceptual spaces? How could their tool support this?

In fact PlaceCells already inherit from a more abstract class (Neurons) which contains basic infrastructure for initialisation, saving data, and plotting data etc. We prefer the solution that users can write their *own* cell classes which inherit from Neurons (or PlaceCells if they wish). Then, users need only write a new get_state() method which can be as simple or as complicated as they like. Here are two examples we’ve already made which can be found on the GitHub:

Phase precession: PhasePrecessingPlaceCells(PlaceCells)^[12]^ inherit from PlaceCells and modulate their firing rate by multiplying it by a phase dependent factor causing them to “phase precess”.

Splitter cells: Perhaps users wish to model PlaceCells that are modulated by recent history of the Agent, for example which arm of a figure-8 maze it just came down. This is observed in hippocampal “splitter cell”. In this demo^[1]^ SplitterCells(PlaceCells) inherit from PlaceCells and modulate their firing rate according to which arm was last travelled along.

R2: – The manuscript also lacks a description of the limitations of the approach. The authors should clarify what types of experimental data that can and cannot be modelled here, the limitations of experimental estimates from the models, and more importantly, what testable predictions of yet unknown results can be generated by this model, beyond replicating what is already known.

We already know that *“testable predictions of yet unknown results can be generated by this model”* as some early citations of the package^[8,9,10]^ (which have been added to the manuscript) demonstrate. However, we think of RatInABox as a generator or models rather than a model itself, therefore making testable predictions is not the primary contribution and might seem out of place. We have modified the following sentence in the discussion to clarify this:

“Its user-friendly API, inbuilt data-plotting functions and general yet modular feature set mean it is well placed empower a wide variety of users to more rapidly build, train and validate models of hippocampal function [25] and spatial navigation[26], accelerating progress in the field.”

As well as:

“In conclusion, while no single approach can be deemed the best, we believe that RatInABox’s unique positioning makes it highly suitable for normative modelling and NeuroAI. We anticipate that it will complement existing toolkits and represent a significant contribution to the computational neuroscience toolbox.”

For example, can RatInABox be used to model episodic coding in the hippocampus (e.g., splitter cells), as task states can often be latent and non-Markovian. Can it be used to model the "remapping" of place cells (and other cell types) in response to changes in the environment?

All this stuff falls well within the scope of RatInABox but we’d rather not go down the rabbit hole of one-by-one including – at the level of user-facing API – all possible biological phenomena of interest. Instead the approach we take with RatInABox (which we think it’s better and more scalable) is to provide a minimal feature set and then *demonstrate* how more complex phenomena can be modelled in case studies, supplementary material and demos.

As you mentioned splitter cells we went ahead and made a very basic simulation of splitter cells which is now available as a Jupyter demo on the GitHub^[1]^. In this demo, the Agent goes around a figure-of-eight maze and “remembers” the last arm it passed through. A bespoke “SplitterPlaceCell” uses this to determine if a splitter cell will fire or not. Of course this is highly simplistic and doesn’t model the development of splitter cells or their neural underpinnings but the point is not to provide a completely comprehensive exploration of the biological mechanisms of splitter cells, or any other phenomena for that matter, but to demonstrate that this is *within scope* and give users starting point. We also have a new demo for making conjunctive grid cells^[6]^ by combining grid cells and head direction cells, and many more. This figure shows a snippet of the splitter cell demo.

Likewise, remapping could absolutely be modelled with RatInABox. In fact we study this in our own recent paper^[8]^ which uses RatInABox. By the way, PlaceCells now contain a naive “remap()” method which shuffles cell locations.

We repeat here the modified paragraph in section 3.1 pointing users to these teaching materials:

“Additional tutorials, not described here but available online, demonstrate how RatInABox can be used to model splitter cells, conjunctive grid cells, biologically plausible path integration, successor features, deep actor-critic RL, whisker cells and more. Despite including these examples we stress that they are not exhaustive. RatInABox provides the framework and primitive classes/functions from which highly advanced simulations such as these can be built.”

Reviewer #2 (Recommendations for the authors):– With a few exceptions, the manuscripts lacks a comparison between the RatInABox and previous approaches to simulate behavioral and electrophysiological data. Such comparison can be added to the Discussion and will help many readers to appreciate the novelty and capabilities of this toolbox.

We have added the following paragraph to the discussion to contrast RatInABox to previous approaches and highlight assumptions/limitations of the package:

“Our package is not the first to model neural data [37, 38, 39] or spatial behaviour [40, 41], yet it distinguishes itself by integrating these two aspects within a unified, lightweight framework. The modelling approach employed by RatInABox involves certain assumptions:

1. It does not engage in the detailed exploration of biophysica l[37, 39] or biochemical[38] aspects of neural modelling, nor does it delve into the mechanical intricacies of joint and muscle modelling[40, 41]. While these elements are crucial in specific scenarios, they demand substantial computational resources and become less pertinent in studies focused on higher-level questions about behaviour and neural representations.

2. A focus of our package is modelling experimental paradigms commonly used to study spatially modulated neural activity and behaviour in rodents. Consequently, environments are currently restricted to being two-dimensional and planar, precluding the exploration of three-dimensional settings. However, in principle, these limitations can be relaxed in the future.

3. RatInABox avoids the oversimplifications commonly found in discrete modelling, predominant in reinforcement learning [22, 23], which we believe impede its relevance to neuroscience.

4. Currently, inputs from different sensory modalities, such as vision or olfaction, are not explicitly considered. Instead, sensory input is represented implicitly through efficient allocentric or egocentric representations. If necessary, one could use the RatInABox API in conjunction with a third-party computer graphics engine to circumvent this limitation.

5. Finally, focus has been given to generating synthetic data from steady-state systems. Hence, by default, agents and neurons do not explicitly include learning, plasticity or adaptation. Nevertheless we have shown that a minimal set of features such as parameterised function-approximator neurons and policy control enable a variety of experience-driven changes in behaviour the cell responses [42, 43] to be modelled within the framework.

In conclusion, while no single approach can be deemed the best, we believe that RatInABox’s unique positioning makes it highly suitable for normative modelling and NeuroAI. We anticipate that it will complement existing toolkits and represent a significant contribution to the computational neuroscience toolbox.”

– An important strength of the toolbox is its capability to simulate with ease realistic animal exploratory patterns. In comparison, the simulation of electrophysiological cell responses is more simplistic. It would be useful to better describe the assumptions and limitations taken in the simulations of cell types and briefly compare them with the well-known experimental evidence. For example, place fields are pre-determined by the model's parameters and completely stable. However, it is well known that place fields developed with experience (e.g., change locations and sizes, pop in/out). The paper claims that it can "concurrently simulate neuronal activity data" in "configurable" environments. It should be clarified if this model can capture the developmental part of place cells, or only the "stable" state.

Assumptions and limitations are described now in the discussion as shown in the section above including a statement clarifying that default cells are steady-state but evolving receptive fields can be modelled too.

RatInABox default cell classes (like PlaceCells, GridCells) are, as you point out, “steady-state”, so don’t by default model adaptation, remapping etc. however these processes can be modelled within the framework quite easily. For example a demo^[4]^ is included which shows how to model successor representations (a popular model for place cells) and shows them “evolving” over time. Another demo shows how to build splitter cells^[1]^ which fire depend on the agents history. Additionally, PlaceCells now have a “remap()” method (released in v1.6.0) which shuffles their receptive fields. All said, we are confident RatInABox can easily extend to modelling learning processes in the brain and non-static receptive fields even if default cell types don’t include these a priori. Minor text changes throughout the manuscript highlight the generality/extensibility of RatInABox and the features which enable this. Additionally the following paragraph in section 3.1 was modified to clarify this:

“Additional tutorials, not described here but available online, demonstrate how RatInABox can be used to model splitter cells, conjunctive grid cells, biologically plausible path integration, successor features, deep actor-critic RL, whisker cells and more. Despite including these examples we stress that they are not exhaustive. RatInABox provides the framework and primitive classes/functions from which highly advanced simulations such as these can be built.”

Finally, RatInABox is open source. This means users can always write their own cell classes which can be arbitrarily complex, time varying etc. This does not merely kick the can down the road. By writing bespoke cells which follow our minimal API they can then immediately benefit from the rest of the RatInABox package functionality (complex Environments, random motion model, plotting functions, multilayer model building etc.). We already know of many users taking this approach of using RatInABox API to construct their own more advanced cell types and we support this.

As another example, the authors showed that the decoding error of Agent position is almost 0 with >= 20 place cells (Figure S1c), which is significantly less than that using real neural data. At least, there should be some discussion of where this difference may arise. Can noise be added to cell tuning curves? E.g. in the form of out-of-field spikes or trial-by-trial variability.

It is absolutely true that this decoding error from our highly simplified analysis is significantly lower than what would be achieved using a comparable number of *real* neurons. There are many reasons for this: real neurons use spikes not rates, real neurons are noisy and neurons may be jointly encoding multiple variables at once, not just position. All of these can be modelled in RatInABox. For example users could extract spikes and train a spike-based decoder. Following your comment noise can now be natively added to all cell types (new as of v1.1.0) as described in a new methods subsection (see 6.3). Mixed selective Neurons can easily be built using function approximator classes as now discussed in the manuscript and demonstrated in a tutorial^[6]^.

As you recommended, we have added the following sentence to the supplementary section to discuss where these differences arise and how they can be studied within the framework.

“Decoding errors in Figure S1c are smaller than would be expected if one decoded from equivalently sized populations of real hippocampal neurons. There are likely many reasons for this. Real neurons are noisy, communicate sparsely through spikes rather than rates and, most likely, jointly encode position and many other behaviourally relevant (or irrelevant) variables simultaneously. All of these factors could be straightforwardly incorporated into this analysis using existing RatInABox functionality.”

On the topic of generality we have been working to make RatInABox more versatile. An illustrative case in point is the new NeuralNetworkNeurons class. These neurons take *other* neurons as inputs and map them through a small (pytorch) neural network embedded within them. Beyond this they follow the standard RatInABox API and can be used exactly as any other cell would be. In a new panel of figure 3 (and an associated demo notebook^[7]^) we demonstrate these by training them to learn a non-linear target function from a set of grid cell inputs. Additionally in a different demo^[3]^ we show how they can support deep reinforcement learning. As always, the point here is not the specifics of the example but to show that with this neurons class (and the analogous FeedForwardLayer) users are not restricted to our predefined list but can in principle create or “learn” arbitrary neural classes with their own complex receptive fields and perhaps even dynamics. This is clarified in the following text in modified/added-to section 2:

Customizable and trainable Neurons: Any single toolkit cannot contain all possible neural representations of interest. Besides, static cell types (e.g. PlaceCells, GridCells etc.) which have fixed receptive fields are limiting if the goal is to study how representations and/or behaviour are learned. RatInABox provides two solutions: Firstly, being open-source, users can write and contribute their own bespoke Neurons (instructions and examples are provided) with arbitrarily complicated rate functions.

Secondly, two types of function-approximator Neurons are provided which map inputs (the firing rate of other Neurons) to outputs (firing rate) through a parameterised function which can be hand-tuned or trained to represent an endless variety of receptive field functions including those which are mixed selective, non-linear, dynamic and non-stationary.

– FeedForwardLayer: Calculates a weighted linear combination of the input Neurons with optional bias and non-linear activation function.

– NeuralNetworkNeurons: Inputs are passed through a user-provided artificial neural network.

Naturally, function-approximator Neurons can be used to model how neural populations in the brain communicate, how neural representations are learned or, in certain cases, neural dynamics. In an online demo we show how grid cells and head direction cells can be easily combined using a FeedForwardLayer to create head-direction selective grid cells (aka. conjunctive grid cells[10]). In Figure 3d and associated demo GridCells provide input to a NeuralNetworkNeuron which is then trained, on data generated during exploration, to have a highly complex and non-linear receptive field. Function-approximator Neurons can themselves be used as inputs to other function-approximator Neurons allowing multi-layer and/or recurrent networks to be constructed and studied.

This panel added to figure 3 demonstrate them in action.

– The manuscript also lacks a description of the limitations of the approach. The authors should clarify what types of experimental data that can and cannot be modelled here, the limitations of experimental estimates from the models, and more importantly, what testable predictions of yet unknown results can be generated by this model, beyond replicating what is already known.

We already know that *“testable predictions of yet unknown results can be generated by this model”* as some early citations of the package^[8,9,10]^ (which have been added to the manuscript) demonstrate. However, we think of RatInABox as a generator or models rather than a model itself, therefore making testable predictions is not the primary contribution and might seem out of place. We have modified the following sentence in the discussion to clarify this:

“Its user-friendly API, inbuilt data-plotting functions and general yet modular feature set mean it is well placed empower a wide variety of users to more rapidly build, train and validate models of hippocampal function [25] and spatial navigation[26], accelerating progress in the field.”

As well as:

“In conclusion, while no single approach can be deemed the best, we believe that RatInABox’s unique positioning makes it highly suitable for normative modelling and NeuroAI. We anticipate that it will complement existing toolkits and represent a significant contribution to the computational neuroscience toolbox.”

For example, can RatInABox be used to model episodic coding in the hippocampus (e.g., splitter cells), as task states can often be latent and non-Markovian. Can it be used to model the "remapping" of place cells (and other cell types) in response to changes in the environment?

All this stuff falls well within the scope of RatInABox but we’d rather not go down the rabbit hole of one-by-one including – at the level of user-facing API – all possible biological phenomena of interest. Instead the approach we take with RatInABox (which we think it’s better and more scalable) is to provide a minimal feature set and then *demonstrate* how more complex phenomena can be modelled in case studies, supplementary material and demos.

As you mentioned splitter cells we went ahead and made a very basic simulation of splitter cells which is now available as a Jupyter demo on the GitHub^[1]^. In this demo, the Agent goes around a figure-of-eight maze and “remembers” the last arm it passed through. A bespoke “SplitterPlaceCell” uses this to determine if a splitter cell will fire or not. Of course this is highly simplistic and doesn’t model the development of splitter cells or their neural underpinnings but the point is not to provide a completely comprehensive exploration of the biological mechanisms of splitter cells, or any other phenomena for that matter, but to demonstrate that this is *within scope* and give users starting point. We also have a new demo for making conjunctive grid cells^[6]^ by combining grid cells and head direction cells, and many more. This figure shows a snippet of the splitter cell demo.

Likewise, remapping could absolutely be modelled with RatInABox. In fact we study this in our own recent paper^[8]^ which uses RatInABox. By the way, PlaceCells now contain a naive “remap()” method which shuffles cell locations.

We repeat here the modified paragraph in section 3.1 pointing users to these teaching materials:

“Additional tutorials, not described here but available online, demonstrate how RatInABox can be used to model splitter cells, conjunctive grid cells, biologically plausible path integration, successor features, deep actor-critic RL, whisker cells and more. Despite including these examples we stress that they are not exhaustive. RatInABox provides the framework and primitive classes/functions from which highly advanced simulations such as these can be built.”